# Endogenous opioids regulate moment-to-moment neuronal communication and excitability

Bryony L. Winters[1], Gabrielle C. Gregoriou[1], Sarah A. Kissiwaa[1], Oliver A. Wells[1], Danashi I. Medagoda[1], Sam M. Hermes[2], Neil T. Burford[3], Andrew Alt[3], Sue A. Aicher[2] & Elena E. Bagley[1]

Fear and emotional learning are modulated by endogenous opioids but the cellular basis for this is unknown. The intercalated cells (ITCs) gate amygdala output and thus regulate the fear response. Here we find endogenous opioids are released by synaptic stimulation to act via two distinct mechanisms within the main ITC cluster. Endogenously released opioids inhibit glutamate release through the δ-opioid receptor (DOR), an effect potentiated by a DOR-positive allosteric modulator. Postsynaptically, the opioids activate a potassium conductance through the μ-opioid receptor (MOR), suggesting for the first time that endogenously released opioids directly regulate neuronal excitability. Ultrastructural localization of endogenous ligands support these functional findings. This study demonstrates a new role for endogenously released opioids as neuromodulators engaged by synaptic activity to regulate moment-to-moment neuronal communication and excitability. These distinct actions through MOR and DOR may underlie the opposing effect of these receptor systems on anxiety and fear.

[1] Discipline of Pharmacology, School of Medical Sciences, University of Sydney, W300 Blackburn Building, Blackburn Circuit, Camperdown, Sydney, New South Wales 2006, Australia. [2] Department of Physiology and Pharmacology, Oregon Health and Science University, 3181 SW Sam Jackson Park Road, Portland, Oregon 97239-3098, USA. [3] Bristol-Myers Squibb, 5 Research Parkway, Wallingford, Connecticut 06492, USA. Correspondence and requests for materials should be addressed to E.E.B. (email: elena.bagley@sydney.edu.au).

The amygdala is central to our emotional responses[1]. In particular, anxiety and fear learning rely on neural circuits and synaptic plasticity within the amygdala[1,2]. These anxiety and fear responses are modulated by our endogenous opioid system. Deleting one family of endogenous opioids, the enkephalins, increases behavioural measures of fear and anxiety[3,4], whereas inhibiting enkephalin breakdown reduces these behaviours[5]. This suggests that endogenous enkephalin is anxiolytic. However, enkephalin is an agonist at both the μ-opioid receptor (MOR) and δ-opioid receptor (DOR)[6,7] and the consequence of activating each receptor results in opposing behavioural outcomes. Indeed activation of DOR is anxiolytic[5,8,9], while activation of MOR is anxiogenic[9,10]. Given this complexity, understanding the cellular actions of endogenous opioids at the DOR and MOR in the amygdala is critical if we hope to utilize opioid related therapy for emotional disorders. However, while endogenous opioids regulate fear and many other behaviours, including: pain, decision making, drug dependence and memory[11], their cellular actions in the brain are poorly understood. Previous studies in various brain regions suggest endogenous opioid regulation of synaptic activity requires intense stimulation[12,13] and often this is shown to regulate long-term plasticity of synapses[12,14–18], rather than normal synaptic transmission. This has been taken to suggest that endogenously released opioids regulate learning rather than continuous information flow through a dynamic neural circuit[19]. However, it is an open question whether endogenously released opioids produce this myriad of behavioural responses solely through the regulation of synapses under intense neuronal activity (such as during learning), or whether regulation under basal conditions also contributes.

Opioid receptors and peptides are expressed to varying degrees throughout the amygdala[20,21]. In particular, the intercalated cells (ITCs) are one possible site where enkephalin could regulate fear and anxiety behaviours. ITCs are small clusters of densely packed GABAergic neurons that en-sheath the basolateral amygdala (BLA). Coronal sections give rise to three separate clusters: the smaller lateral (lpc) and medial (mpc) paracapsular ITC clusters are located within the external and intermediate capsules respectively and the larger main island (Im), is located ventromedial to the BLA[22] (Fig. 1a). While the lpc provides feedforward inhibition to the BLA[23], the mpc acts as an inhibitory interface between the BLA and CeA and thus regulates fear learning[24]. In particular the mpcs are required for fear extinction[25]. Less is known about the functional role of the main island although it is possible the Im plays a similar role to the mpc. Indeed, Im neurons also receive sensory information from both the BLA[24,26] and the thalamus[27] along with more complex information from the medial pre-frontal cortex (mPFC), a region highly implicated in fear extinction[28,29]. Like the mpcs, the Im sends inhibitory GABAergic projections to the medial central nucleus (CeM)[21,26,30] and thus could gate expression of the conditioned fear response[31]. The Im may be particularly important during fear extinction as extinction activates Im neurons[22,32], their ablation (along with the mpc) reduces extinction[25] and treatments that reverse the extinction deficit in anxious mice elevate Im neuron activity[32].

Im neurons strongly expresses enkephalin[20,21,33] that could act at pre- or postsynaptic sites. For example, the glutamatergic synaptic input from the BLA could be targeted by endogenously released opioids, as BLA pyramidal neurons express both MOR and DOR[21,34]. Further, in other subdivisions of the amygdala, at least 60% of MOR is found in postsynaptic dendritic compartments, rather than in axons or terminals, suggesting potential for postsynaptic regulation[34,35]. To date, endogenously released opioids have not been shown to directly activate a postsynaptic conductance in any brain region. However, given Im neurons express high levels of MOR[21] and exogenously applied MOR agonists activate a potassium conductance in Im neurons (Im referred to as medial ventral ITC in this study)[16] it is feasible Im neurons may be postsynaptically regulated by endogenously released opioids.

Using electrophysiology combined with immunohistochemistry, we find that endogenously released opioids directly activate a potassium conductance in Im neurons via MOR. We also find that whilst exogenous Met-enkephalin (ME) and selective MOR, and DOR agonists reduced glutamate release from BLA synaptic inputs, endogenously released opioids only inhibit glutamate release through DOR. These endogenous opioid actions could be potentiated by reducing endogenous opioid breakdown by peptidases or by a positive allosteric modulator (PAM) specific for DOR. These findings indicate that endogenously released opioids tightly regulate the excitability and synaptic activation of Im neurons through two different receptors. Further, this regulation is not limited to high intensity synaptic activity, suggesting a role for endogenously released opioids as regulators of moment-to-moment signalling within a dynamic neuronal network. This would be expected to contribute to the role of endogenous opioids in fear learning and anxiety.

## Results

### ME in dense core vesicles overlaps with MOR expression in Im.
Consistent with previous reports, we found high immunoreactivity for both MOR and ME in the main ITC cluster (Im; Fig. 1b)[20,21]. Strong ME immunoreactivity also occurred in the central nucleus of the amygdala (CeA) and in a scattering of cells within the BLA (Fig. 1b). Diffuse MOR immunoreactivity was found in both the BLA and CEA (Fig. 1b).

Given the overlapping distribution of ME and MOR within Im (Fig. 1b, merge) we focused specifically on this cluster of ITCs for this study. To explore possible ME targets, we used immuno-electron microscopy to determine the ultrastructural location of ME within the Im region. We found that ME immunoreactivity was concentrated in dense core vesicles (DCVs) located in axons and axon terminals (Fig. 1c–e). These terminals were directly apposed to (Fig. 1c,d) or convergent with (Fig. 1e) axon terminals that form asymmetrical (assumed glutamatergic) synapses onto dendrites. Therefore, ME is well placed to be released following synaptic activation and in turn, modulate glutamate release within Im.

### DOR and MOR agonists reduce glutamate release from the BLA.
One of the main glutamatergic inputs to Im neurons is from pyramidal neurons of the BLA. BLA-Im synapses (Im referred to as medial ventral ITC in ref. 16) in mice have recently been shown to be insensitive to regulation by MOR[16], however, it is not known whether this holds across different species. We therefore tested the sensitivity of this synapse to all three opioid receptors. To do this we made whole-cell patch recordings from Im neurons and routinely performed *post hoc* biocytin staining of the patched cells. As expected, staining revealed small bipolar neurons, with an abundance of dendritic spines, located in a cell dense region ventromedial to the BLA (Fig. 2a)[26,36]. We electrically stimulated the BLA (Fig. 2b) and recorded the resulting evoked excitatory postsynaptic current (eEPSC) in Im neurons. We found BLA-Im eEPSCs were inhibited by selective agonists for DOR (Deltorphin II, 300 nM) or MOR (DAMGO, 1 μM; Fig. 2c,d), which was accompanied with an increase in the paired pulse ratio (PPR, Fig. 2e). Both effects were fully reversed by the selective DOR (ICI174864, 1 μM) and MOR (CTAP, 1 μM) antagonists respectively (Fig. 2c–e). In contrast, neither the

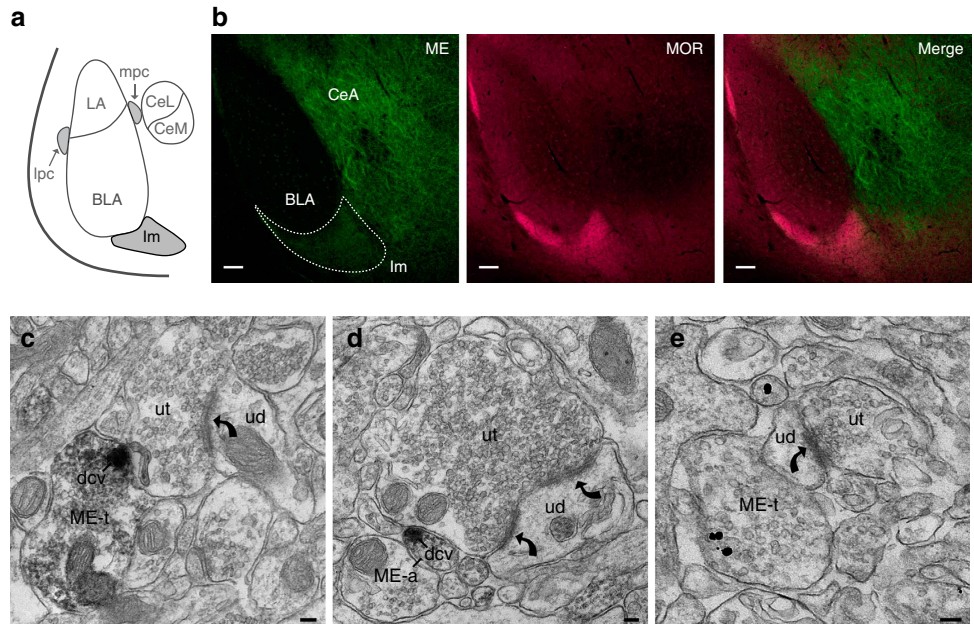

**Figure 1 | Met-enkephalin expressed within the main ITC island is positioned to modulate excitatory synapses.** (**a**) Schematic of amygdala subdivisions. ITC islands are shaded in grey. BLA, basolateral amygdala; CeL, lateral central amygdala and CeM, medial central amygdala make up the central amygdala (CeA); Im, Main ITC island; lpc, lateral ITC; mpc, medial ITC. (**b**) Single confocal images in the amygdala (Bregma − 2.00 mm) of ME (green), MOR (magenta) and the two merged channels. Im is outlined by the dashed line. Scale bars, 100 μm. (**c–e**) electron micrographs showing: (**c**) ME immunoreactivity is located in an axon terminal (ME-t) and concentrated in a dense core vesicle (dcv) that is directly apposed to an unlabeled terminal (ut) that forms an asymmetric synapse (curved arrow) with an unlabelled dendrite (UD). (**d**) An, unmyelinated axon (ME-a) contains a dense core vesicle that is immunoreactive for ME and is apposed to an unlabeled axon terminal (ut) that forms a perforated asymmetric synapse (curved arrows) with an unlabelled dendrite (UD). (**e**) ME immunogold immunoreactivity is located in an axon terminal (ME-t) apposed to an unlabelled dendrite (ud) that receives an asymmetric synapse (curved arrow) from an unlabelled terminal (ut). Scale bars, 100 nm.

κ-opioid receptor (KOR) agonist (U69593, 3 μM) nor antagonist (norBNI, 10 nM) affected the eEPSC amplitude or PPR (Fig. 2c–e). These data indicate the BLA-Im synapse is strongly inhibited through DOR and MOR and the associated increases in PPR indicates the likely site of action is through pre-synaptic reductions in glutamate release. This opioid sensitivity is consistent with the high expression of DOR mRNA in BLA neurons[21] and the expression of MOR in BLA pyramidal neurons[34] but does differ from the recently reported MOR insensitivity of this synapse described in mice[16], suggesting there may be inter-species variability in opioid action within the Im.

ME is a high-affinity ligand for both MOR and DOR[6,7] and is therefore likely to inhibit the BLA-Im synapse. Exogenous ME (10–30 μM) significantly inhibited the eEPSC amplitude (Fig. 2f,g) and increased the PPR (Fig. 2h). Both DOR and MOR selective antagonists produced a partial reversal of ME inhibition and together they fully reversed both the ME induced inhibition (Fig. 2i) and change in PPR (baseline: 1.3 ± 0.1; ME: 1.8 ± 0.2; ICI + CTAP: 1.2 ± 0.2; n = 8, P > 0.05 baseline versus ICI + CTAP, paired t-test). Unsurprisingly, given the low affinity of ME for KOR[6,7] we found norBNI had no effect on the ME inhibition (Fig. 2i). Thus, exogenous ME inhibited glutamate release at the BLA-Im synapse through activation of both MOR and DOR, which is consistent with the receptor pharmacology of ME[6,7] and the opioid sensitivity of the BLA-Im synapse, as defined above.

**Endogenously released opioids inhibit synaptic activity.** In other brain regions, detecting the actions of endogenously released opioids requires intense stimulation[12–18]. Consistent with this, in the hypothalamus, where dense core vesicle release

has been most thoroughly studied, optimal release is elicited by intense stimulation (at least 5–10 Hz)[37]. However, less intense stimulation (1 Hz) can elicit small but significant peptide release[37]. In light of this, we tested whether we could stimulate endogenous opioid peptide release using less intense stimulation. We initially tested whether the standard paired-pulse stimulation ('low stimulus', Fig. 3a) was sufficient to produce an endogenous opioid effect. To determine the actions of endogenously released opioids, we examined whether the opioid antagonist naloxone (10 μM) increased the eEPSC amplitude, with an increase taken to indicate a reversal of endogenous opioid inhibition (Fig. 3a). However, naloxone did not increase eEPSC amplitude at the BLA-Im synapse in response to the low-stimulus protocol (Fig. 3b,i).

In the striatum, intense stimulation (five antidromic depolarizations, 100 Hz) results in opioid inhibition of glutamate release that is maximal 500 ms after the stimulus[13]. Guided by this, we delivered a short train of stimuli ('moderate' stimulus, 5 stimuli, 150 Hz) followed by a single 'test' stimulus (500 ms interval) to the BLA (Fig. 3a). Unless otherwise noted, we analysed opioid effects on the test eEPSC. Using this protocol, we found that naloxone significantly increased the amplitude of the test eEPSC (Fig. 3c,i), indicating the moderate stimulus induces endogenous opioid release. We wondered whether opioid release occurs with lower stimulation but due to peptide degradation, we were unable to detect the opioid effect. To test this, we assessed whether inhibition of opioid peptide degradation could reveal an opioid effect during low stimulation. Enkephalin, the most likely opioid to be acting at this synapse, is catabolized by at least three zinc metalloproteases[38]. We tested whether a cocktail of peptidase inhibitors targeting these enzymes (thiorphan, 10 μM; captopril, 1 μM and bestatin, 10 μM; Fig. 3d), could increase endogenous

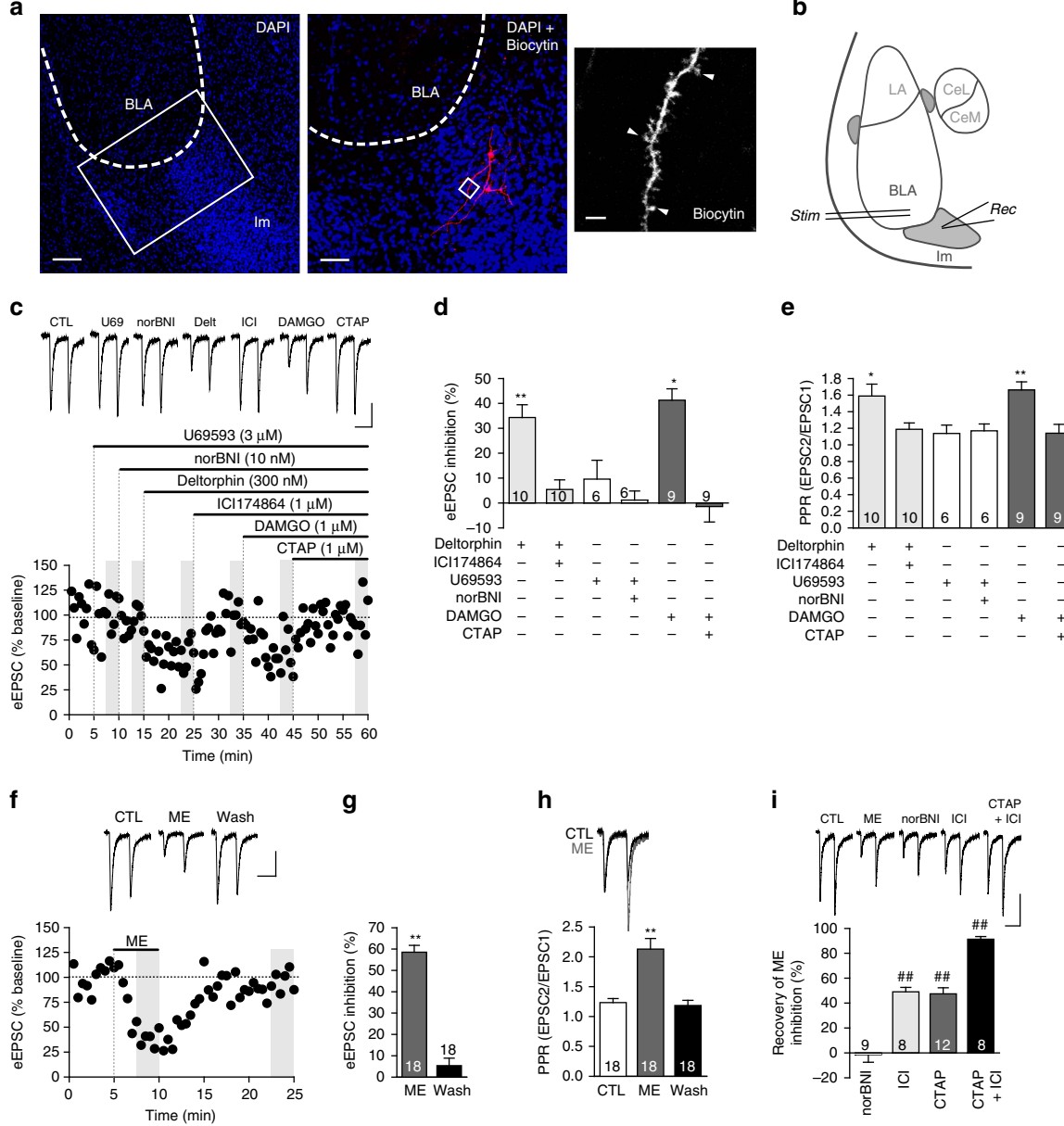

**Figure 2 | DOR and MOR activation inhibits glutamate release at the BLA-ITC synapse.** (**a**) *Post hoc* confocal images of DAPI (blue) and biocytin-labelled ITCs (red/white). Low power image shows intense DAPI labelling with high cell density within Im. Single image; scale bar, 100 μm. Magnification ( × 20 objective) of boxed area, shows two filled ITCs with bipolar characteristics. Stack image ($z = 13.5$ μm); scale bar, 50 μm. Further magnification ( × 63, boxed area) reveals dendritic spines (arrows); single $Z$ section ($z = 2.5$ μm); scale bar, 3 μm. (**b**) Representative BLA-Im *stim*ulation and *rec*ording locations. (**c–e**) Selective DOR (deltorphin II, Delt 300 nM) and MOR (DAMGO, 1 μM)) agonists, reduced eEPSC amplitude and increased PPR that was reversed by the corresponding antagonists ICI1173864 (ICI, 1 μM) and CTAP (1 μM), respectively. KOR agonist (U69, 3 μM) and antagonist norBNI (10 nM), had no effect. (**c**) Example eEPSCs and time plot of normalized peak amplitude (eEPSC$_1$) of a single representative experiment. (**d**) Bar chart showing percentage inhibition from baseline. Baseline defined as eEPSC amplitude of CTL (no drug) or previous antagonist. (**e**) Bar chart showing PPR, calculated as eEPSC$_2$/eEPSC$_1$. (**f–h**) ME (10 μM) reduced eEPSC amplitude and increased PPR which fully reversed after wash. (**f**) Example eEPSCs and time plot of single representative experiment; (**g**) bar chart of mean inhibition during ME and wash. (**h**) Bar chart of mean PPR (bottom) and example traces (top) normalized to first eEPSC before (black) and during ME (grey). (**i**) ME inhibition was fully reversed by combined DOR and MOR antagonist treatment. Example eEPSCs and bar chart showing the proportion of ME inhibition of eEPSC amplitude (%) reversed by the selective antagonists. (Data are represented as mean ± s.e.m.; *$P < 0.05$, **$P < 0.01$, paired *t*-test, from CTL; ##$P < 0.01$, paired *t*-test versus ME.) Highlighted regions on time plots represent region sampled for bar charts. Scale bars, 50 ms, 100 pA.

opioid inhibition. This peptidase inhibitor cocktail significantly enhanced the effects of exogenously applied, submaximal ME (300–500 nM) on eEPSC amplitude (Fig. 3e,f), confirming these peptidases act to break down exogenous ME and thus limit its actions at the BLA-Im synapse.

It is important to note these targeted peptidases are non-selective metalloproteases. Therefore, blocking their activity has

the potential to enhance the activity of other 'off-target' endogenous signalling peptides (for example, substance P[39,40], neurokinin[41,42]; neurotensin[42,43]) that may act either in concert or opposition to endogenous opioid signalling. Consistent with this, while inhibiting peptidase activity with the peptidase inhibitor cocktail on average, reduced eEPSC amplitude in response to both low (12.9 ± 5.4% inhibition, $n = 7$) and

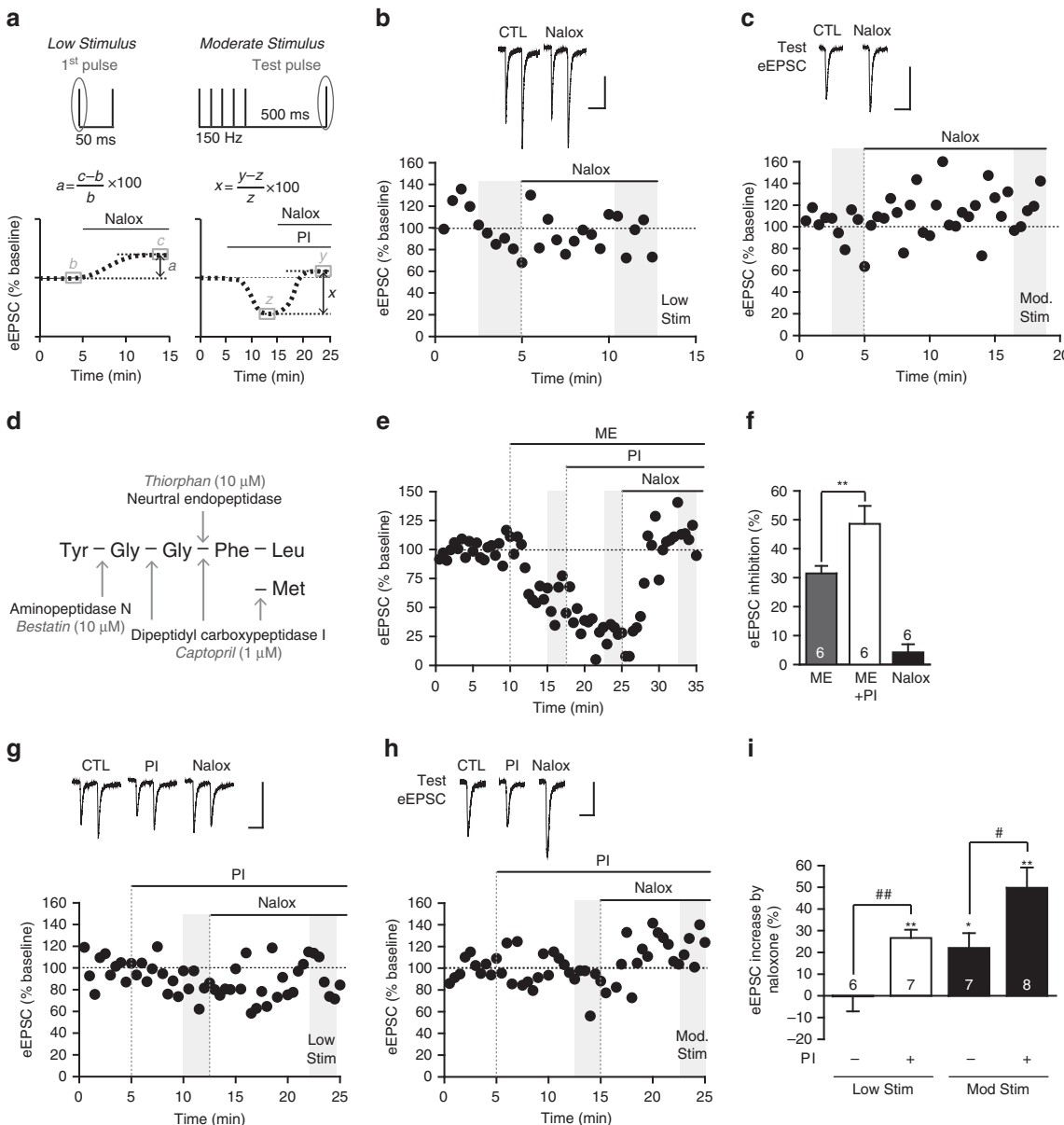

**Figure 3 | Endogenously released opioids inhibit glutamate release at the BLA-ITC synapse. (a)** Graphic showing stimulus paradigms and calculation of endogenous opioid effect. **(b)** Naloxone (Nalox, 10 μM) has no effect on eEPSC amplitude when evoked with a low stimulus, **(c)** while amplitude of the moderate stimulus 'test eEPSC' is increased by naloxone. Data displaying eEPSCs and time plots of single representative experiments. **(d)** Met/leu-enkephalin is catabolized by specific peptidases; diagram showing cleavage sites, peptidases and their corresponding inhibitors (PIs). **(e,f)** Inhibition of eEPSC amplitude with submaximal ME (300–500 nM) is potentiated by peptidase inhibitors (PI; thiorphan 10 μM, captopril 1 μM, bestatin 10 μM). **(e)** Representative time plot and **(f)** bar chart showing percentage inhibition of eEPSC amplitude (from baseline). When peptidases are inhibited, naloxone now **(g)** increases synaptic responses evoked with the low stimulus and **(h)** produces a larger increase in amplitude of the moderate stimulus test eEPSC. Data displaying eEPSCs and time plots of single representative experiments. **(i)** The increase in synaptic response by naloxone increases with greater stimulus intensity and/or inclusion of PIs. Bar chart shows mean increase by naloxone (%) across each condition (Data are represented as mean ± s.e.m.; *P < 0.05, **P < 0.01, paired t-tests, versus CTL; #P < 0.05, ##P < 0.01, unpaired t-tests as indicated). Highlighted regions on time plots represent region sampled for bar charts. Scale bars, 50 ms, 100 pA.

moderate (32.1 ± 6.2% inhibition, n = 8) stimuli (for example, Fig. 3g,h), there was a high degree of variability, with either inhibition (12/18 neurons); no effect (3/18 neurons) or an increase observed (3/18 neurons). Thus, we defined endogenous opioid action as the naloxone-induced increase in eEPSC following peptidase inhibitor treatment (Fig. 3a). In addition, there was variability in the timing of the peptidase inhibitor response, which could reflect the time required to accumulate sufficient peptide or differences in restricting microarchitecture. Despite this variability in peptidase inhibitor response, naloxone

significantly increased eEPSC amplitudes in all cells. Due to possible dominance of confounding 'off-target' signalling peptides in cells where the peptidase inhibitors increased the eEPSC amplitude, these cells were excluded from further analysis of the endogenous opioid effect. In the remaining cells, naloxone induced a significant increase in eEPSC amplitude using both low (Fig. 3g,i) and moderate stimuli (Fig. 3h,i). These data indicate endogenous opioids are released during low-stimulus experiments but eEPSC amplitude remains unchanged due to rapid degradation of the peptide. Further, it shows that the

actions of endogenously released opioids, released in response to the moderate stimulus, are limited by peptidase activity. Interestingly, when peptidases are inhibited, the first pulse of the moderate stimulus train was also increased by naloxone ($40.2 \pm 4.8\%$ increase, $n = 8$, $P < 0.01$ peptidase inhibitor versus naloxone, paired $t$-test). As this did not occur without peptidase inhibitors ($10.7 \pm 6.4\%$ increase, $n = 7$, $P > 0.05$ baseline versus naloxone, paired $t$-test), this suggests endogenously released opioids are broken down by peptidases during the 15 s inter-stimulus interval.

**Endogenously released opioids signal through DOR.** We tested whether endogenously released opioids acted through DOR or MOR using our strongest paradigm (moderate stimulus and peptidase inhibitor cocktail), and applying selective DOR and MOR antagonists. We found that while CTAP had no effect, ICI174864 increased eEPSC amplitude (Fig. 4a,b) to the same extent as naloxone ($P > 0.05$, unpaired $t$-test). While ICI174864 does have modest inverse agonist activity at DOR, since the effect of ICI174864 was not different from the neutral antagonist naloxone, inverse agonism is unlikely to explain the current findings[44]. These data indicate that endogenously released opioids only inhibit the BLA-Im synapse through DOR activation.

**DOR-PAM enhances endogenous effect.** PAMs of DOR could potentiate the inhibition by endogenously released opioids. We explored this possibility using the DOR PAM BMS-986187, which shows 100-fold selectivity for DOR over MOR and increases efficacy, potency and affinity of orthosteric ligands[45]. BMS-986187 ($1 \mu M$) significantly enhanced the inhibition induced by submaximal, exogenous ME (100 nM, Fig. 4c,e), indicating PAM activity at this synapse, which could be reversed to baseline levels by naloxone ($6.3 \pm 2.6\%$ inhibition; $P > 0.05$ naloxone versus baseline, Tukey's *post hoc* comparisons, Fig. 4c,e). BMS-986187 is known to have agonist properties at higher concentrations[45] and BMS-986187 alone induced a small but significant decrease in eEPSC amplitude (Fig. 4d,e; $P < 0.05$, paired $t$-test). In these experiments, we used single synaptic stimuli (0.06 Hz) to minimize release of endogenous opioids. However, this BMS-986187 inhibition was reversed by naloxone (Fig. 4d,e) and we found that direct agonism by BMS-986187, resulting in β-arrestin recruitment and inhibition of cAMP production in CHO-OPRD1 cells, was insensitive to naloxone (Supplementary Fig. 1). Therefore, the naloxone sensitive inhibition of the BLA-Im synapse by BMS-986187 alone is not due to intrinsic agonist activity of BMS-986187, rather it is likely due to positive allosteric modulation of endogenously released opioids signalling at DORs. Although we know that PAMs can increase responses to exogenous opioids in cell lines[45], this is the first instance in which increases in endogenous opioid signalling have been observed (Fig. 4d,e). To further investigate PAM activity on endogenous opioid signalling, we tested the effects of BMS-986187 on our previous treatment paradigms. When slices were preincubated with BMS-986187 ($> 45$ min) and peptidases were inhibited, the naloxone-induced increase was significantly greater than peptidase inhibitors alone in response to the low stimulus (Fig. 4f,g), but had no further effect when using the moderate stimulus (CTL: $36.0 \pm 6.5\%$ increase, $n = 8$; BMS-986187: $36.8 \pm 5.9\%$ increase $n = 5$, $P > 0.05$, unpaired $t$-test). Since naloxone produced comparable increases in eEPSC irrespective of stimulation intensity when in the presence of BMS-986187, this may indicate saturation of endogenously released opioid regulation at this synapse.

**Endogenous opioids activate a postsynaptic conductance.** Exogenous opioids activate a potassium conductance in Im neurons[16] and in many other brain regions[46,47]. However, it is not known whether endogenously released opioids can activate this potassium conductance. Given the strong endogenous opioid regulation in Im neurons (current findings), their very high MOR expression[21] and that 60% of MOR receptors are in postsynaptic dendritic compartments[34,35], we wondered whether endogenously released opioids could directly regulate Im neurons. We have already shown ME is well placed to regulate synaptic transmission (Fig. 1c–e), but we also found ME immunoreactivity in multiple terminals converging onto large dendrites (Fig. 5a) within Im, suggesting ME directly acts at postsynaptic sites. Indeed, exogenous ME ($30 \mu M$) produced an outward current in all neurons ($37.5 \pm 4.1$ pA, $n = 20$) that could be readily reversed either after washout ($n = 16$) or by CTAP ($n = 4$; Fig. 5b). Current–voltage analysis before and during ME indicates activation of a potassium conductance as the reversal potential of the ME-induced current was close to the potassium reversal potential ($ME_{rev}$: $-103.2 \pm 1.4$ mV, $n = 9$; $E_k$: $-104.9$ mV, calculated with the Nernst equation; Fig. 5c). In addition, bath application of the $K_{IR}$ 3.1/3.4 antagonist tertiapin Q (300 nM) for 5 min prior to application of ME significantly reduced the ME current (ME + tertiapin $7.4 \pm 1$ pA, $n = 5$, ME alone $37.5 \pm 4.1$ pA, $n = 20$; $P < 0.002$, unpaired $t$-test). This is consistent with findings from mouse Im neurons where the MOR agonist DAMGO activated a potassium conductance and induced a hyperpolarization[16]. To test whether endogenously released opioids also activate this potassium conductance in Im neurons, we applied the peptidase inhibitor cocktail and observed an outward current in the majority of Im neurons (8/9), which was fully reversed by CTAP ($n = 4$) or naloxone ($n = 5$, Fig. 5d,f). While these peptidase inhibitor-induced currents occurred without any electrical stimulation, the highly variable amplitude (Fig. 5d,f) suggested spontaneous activity of cells in the slice could be influencing the size of the endogenous opioid response. Consistent with this, we found that when we electrically stimulated the BLA (10–20 stimuli at 150 Hz), PIs induced a larger, more consistent outward current, which again was fully reversed by CTAP ($n = 1$) or naloxone ($n = 3$; Fig. 5e,f). These data indicate that in the Im, endogenous opioids are readily released in response to spontaneous or stimulated synaptic activity and act postsynaptically to induce an outward potassium conductance. Given the input resistance of Im neurons ($643.9 \pm 101.5$ MΩ, $n = 13$), the endogenous opioid current with and without stimulation would be expected to hyperpolarize the Im neurons by $24.2 \pm 2.7$ and $10.1 \pm 3.8$ mV, respectively.

**Endogenously released opioids inhibit local Im synapses.** ITCs have high intrinsic connectivity within the Im[22,26] and it is possible endogenously released opioids could also inhibit these GABAergic synapses. To study Im-Im neuron synapses exclusively, excluding GABAergic inputs from other regions such as the CeA, we performed paired recordings between synaptically coupled Im neurons (Fig. 6a). We found that exogenous ME ($10 \mu M$) strongly inhibited paired IPSCs (Fig. 6b,c) and in three of the four synaptic pairs, ME also increased synaptic failure rate (CTL: $3.75 \pm 0.1\%$ failure, ME: $45.6 \pm 0.4\%$ failure, for example, Fig. 6b). This inhibition was reversed by either CTAP (Fig. 6b) or wash of ME and likely resulted from ME acting pre-synaptically to reduce GABA release as ME significantly increased the PPR (Fig. 6d). Unfortunately, the success rate for obtaining viable synaptically coupled pairs of Im neurons was low (2.4%) and as a result, we were unable to test the actions of endogenously released opioids using this approach.

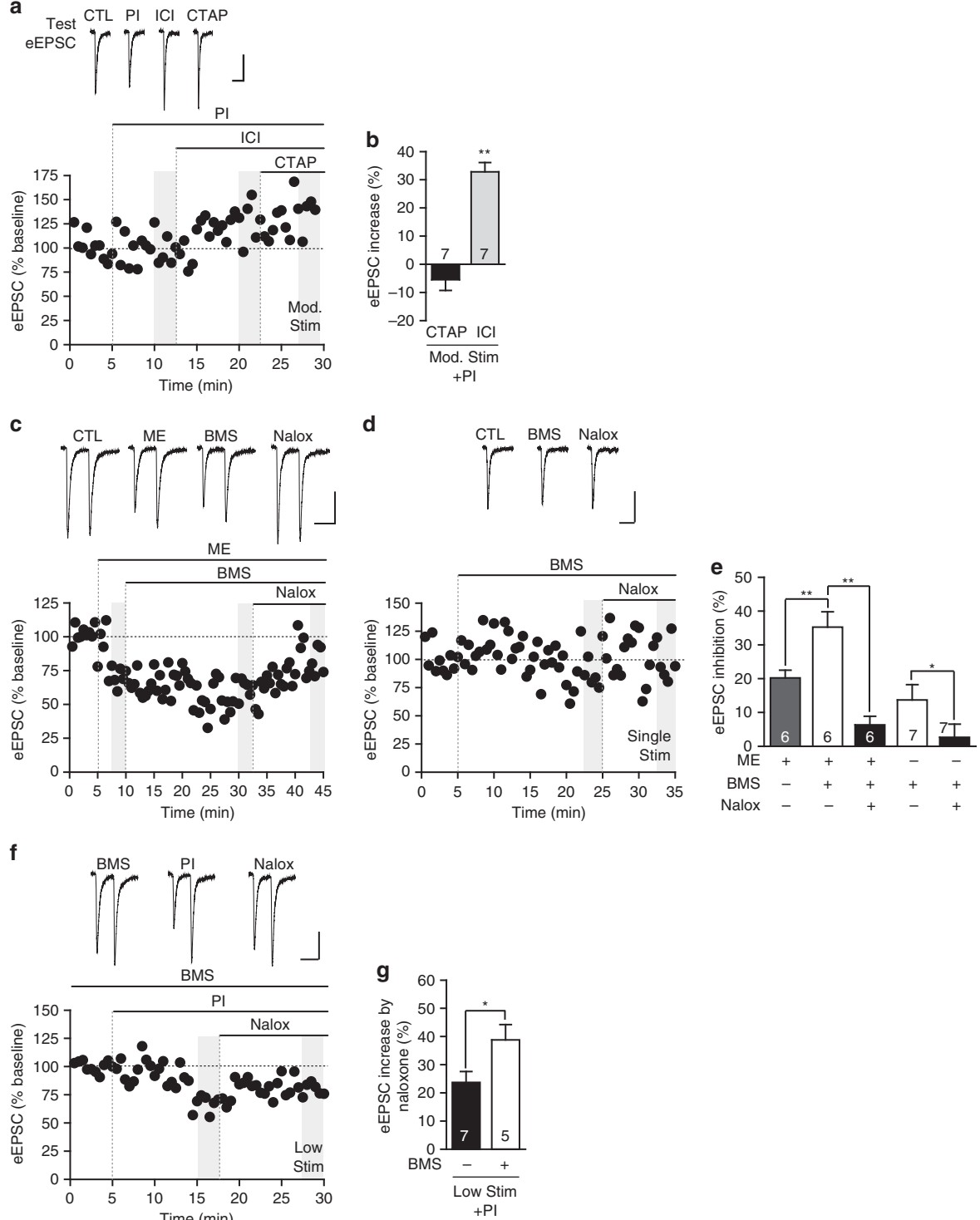

**Figure 4 | Endogenously released opioids act exclusively through DOR and their signalling is enhanced by a DOR PAM. (a,b)** Endogenous opioids signal exclusively through DOR. **(a)** Moderate stimulus 'test eEPSCs' and corresponding time plot from single representative experiment. **(b)** Bar chart of percentage increase of eEPSC amplitude by selective antagonists. MOR antagonist CTAP (1 μM) produced no change while DOR antagonist, ICI174864 (ICI, 1 μM) significantly increased test eEPSC amplitude (**$P < 0.01$, paired $t$-test, versus PI). **(c–e)** The DOR PAM BMS-986187 (BMS, 1 μM) **(c)** significantly enhanced exogenous ME inhibition (submaximal, 100 nM) after at least 15 min continuous perfusion and **(d)** decreased single stimulus evoked eEPSC amplitude in absence of exogenous ligand, both were fully reversed by naloxone indicating PAM and not agonist activity of BMS-986187 (see also Supplementary Fig. 1). **(c,d)** Data displaying eEPSCs and time plots of single representative experiment and **(e)** summary bar chart (**$P < 0.01$, repeated ANOVA; *$P < 0.05$, paired $t$-test). **(f,g)** Preincubation (>45 min) and subsequent continual perfusion of BMS-986187 significantly enhanced endogenous opioid signalling under 'low stimulus' conditions, in the presence of PIs. **(f)** eEPSCs and time plot of single representative experiment. **(g)** Summary bar chart showing BMS-986187 significantly enhanced the naloxone-induced increase of eEPSC amplitude (**$P < 0.01$, unpaired $t$-test). Data are represented as mean ± s.e.m. Highlighted regions on time plots represent region sampled for bar charts. Scale bars, 50 ms, 100 pA.

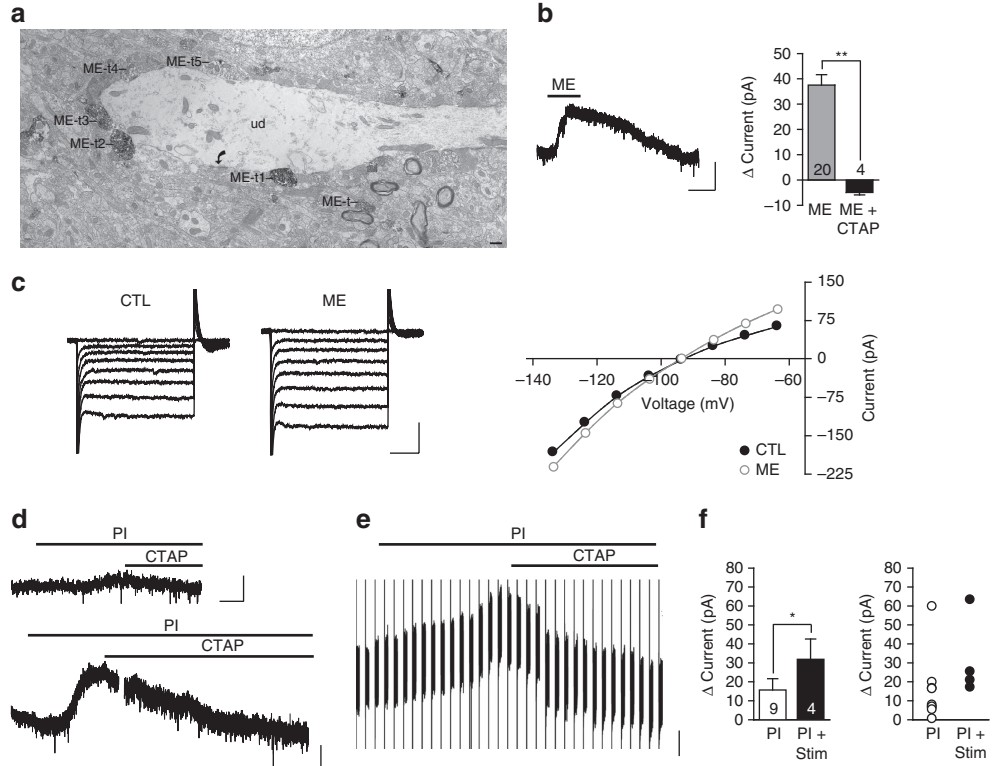

**Figure 5 | Endogenously released opioids activate a postsynaptic conductance in ITC neurons.** (**a**) ME-ir axon terminals converge onto a single postsynaptic target. Numerous ME terminals (MEt1–5) contact an unlabeled dendrite (ud). ME immunoperoxidase reactivity was observed in axon terminals across a spectrum of density and localization ranging from densely filling the entire terminal (ME-t1–3), to diffuse localization within a terminal (MEt4), to discrete compartmentalization in dense core vesicles (ME-t5). ME-ir terminals (ME-t) were also observed elsewhere in the field. Curved arrow indicates an asymmetric synapse. Scale bar, 500 nm. (**b**) ME rapidly activates an outward current in ITC neurons through MOR. Example current trace showing ME (30 μM) and rapid washout. Bar chart showing outward current amplitude is blocked by CTAP (1 μM, *$P < 0.05$, paired $t$-test). (**c**) Example current traces and I–V relationship of single representative neuron. Currents produced by voltage steps (10 mV increments) from $-63.6$ to $133.6$ mV before (CTL; left) and during ME (30 μM, right). Reversal potential for the ME-induced current is the point at which the control and ME I-V relationship curves intersect ($-101.6$ mV, this example). (**d–f**) Peptidase inhibition activates an endogenous opioid outward conductance that is increased by slice stimulation. (**d**) Example traces showing peptidase inhibitors (PI) induce a range of outward currents that are reversed by CTAP ($n = 4$) or naloxone ($n = 5$; top: small current; bottom: larger current). (**e**) Example trace showing effect of peptidase inhibitors on consecutive 1.5 s recordings following trains of stimuli (10–20, 150 Hz) delivered every 15 s, upward and downward deflections are the response to stimulation and are followed by holding current recording, gap represents 13.5 s between each stimuli. (**f**) Bar chart and scatter plot showing the amplitude of the outward current with/without stimulation; *$P < 0.05$, Mann–Whitney $U$-test. Neurons voltage clamped at $-64$ mV. Data are represented as mean ± s.e.m. Scale bars, (**b,d**) 1 min, 20 pA; (**c**) 50 ms, 100 pA; (**e**) 20 pA.

Instead, we recorded local Im synaptic activity by electrically stimulating within the Im to evoke eIPSCs (Fig. 6e). Exogenous ME (10–30 μM) robustly inhibited eIPSCs (Fig. 6f,g) and this was reversed by the MOR antagonist CTAP, but not by DOR or KOR antagonists (Fig. 6h). Surprisingly however, we did not find a consistent change in PPR (Fig. 6i), which was in distinct contrast to opioid regulation of glutamatergic inputs (Fig. 2e) and paired Im neuron recordings (Fig. 6a–d). Although unexpected, we reasoned that since ME almost completely eliminated paired Im-Im inputs (Fig. 6b,c), this may allow MOR insensitive GABAergic inputs with different synaptic characteristics to dominate the remaining eIPSC. Indeed, this is supported by the change in eIPSC kinetics before and after ME treatment, in which both the 10–90% rise time (baseline: $1.7 \pm 0.2$ ms; ME: $1.3 \pm 0.1$ ms, $P < 0.01$, paired $t$-test) and the weighted decay time constant ($\tau_w$; baseline: $34.3 \pm 4.8$ ms; ME: $17.8 \pm 2.8$ ms, $P < 0.01$, paired $t$-test) showed a significant decrease.

To test whether endogenously released opioids also inhibit local GABA synapses, we used either the low or moderate stimulus protocol together with the peptidase inhibitor cocktail. When we used the moderate stimulus, peptidase inhibitors

significantly decreased eIPSC amplitude of the test pulse ($13.1 \pm 5.2\%$ inhibition, $P < 0.05$ versus baseline, paired $t$-test, for example, Fig. 6j), which was then significantly increased by naloxone (Fig. 6j,k). However, when the low stimulus was used, neither the peptidase inhibitor cocktail ($10.1 \pm 6.0\%$ inhibition, $n = 6$, $P > 0.05$ versus baseline, paired $t$-test) nor naloxone (Fig. 6k) significantly changed eIPSC amplitude. Thus local GABAergic synapses within Im are inhibited by endogenously released opioids but only with a combination of PIs and the moderate stimulus and even then, only to a modest extent. This is somewhat surprising considering the strong regulation of these synapses by exogenous ME. In fact, while endogenously released opioids inhibit the BLA-ITC synapse to almost 50% of the maximal opioid inhibition of this synapse, at local ITC GABAergic synapses, it is <15%. This difference could be due to less effective stimulation of endogenous opioid release when studying GABAergic synapses and may indicate a requirement for intact glutamatergic signalling. Alternatively, this difference may result from other ultrastructural differences such as receptor location relative to dense core vesicle release sites.

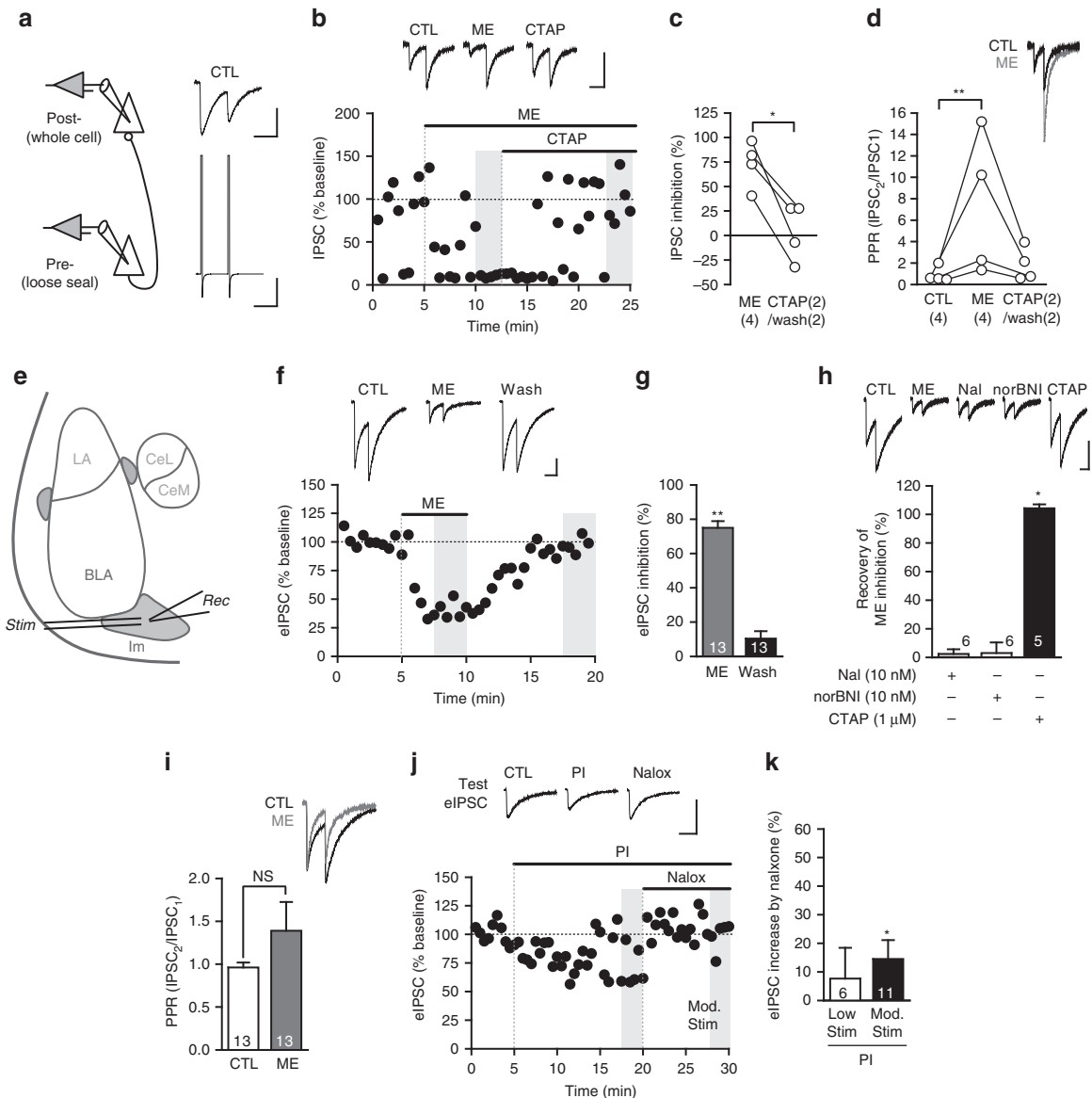

**Figure 6 | Endogenously released opioids modestly inhibit local GABA ITC synapses.** (**a**) Paired recordings of synaptically connected ITCs. Example *pre*-synaptic (lower trace) and *post*synaptic (upper trace) currents following voltage pulses (200 mV) to the pre-synaptic cell. (**b–d**) ME (10 μM) strongly inhibits synaptic transmission between paired ITCs and increases PPR, both are reversed by washout or CTAP. Data showing (**b**) IPSCs and time plot of single representative experiment and (**c**) scatter plot (*$P<0.05$, Friedman's with Fisher's *post hoc* tests). (**d**) Scatter plot of PPR and example IPSCs normalized to first peak (**$P<0.01$, Friedman's with Fisher's *post hoc* tests). (**e**) Stimulation/recording locations for local GABAergic-ITC synapses. (**f,g**) ME (10 μM) strongly inhibits local eIPSCs amplitude, (**f**) eIPSCs and time plot of single representative experiment, (**g**) summary bar chart (**$P<0.01$, paired *t*-test versus CTL). (**h**) ME inhibition is mediated exclusively by MOR. Example eIPSCs and bar chart showing percent recovery from ME inhibition by application of DOR (Naltrindole, 10 nM), KOR (norBNI) and MOR (CTAP) selective antagonists (*$P<0.05$, paired *t*-test, CTAP versus ME). (**i**) ME has no effect on PPR and quickens synaptic decay kinetics; example eIPSCs normalized to first peak showing no change in PPR but faster synaptic decay and summary bar chart of PPR (NS, not significant, paired *t*-test). (**j,k**) In the presence of peptidase inhibitors (PI), naloxone only increased moderate stimulus evoked test eIPSCs. (**j**) Test eIPSCs and time plot of single representative experiment. (**k**) Bar chart showing modest increase in test eIPSC by naloxone but no change in low stimulus eIPSCs. (*$P<0.05$, paired *t*-test versus PI). Data are represented as mean ± s.e.m. Highlighted regions on time plot represents region sampled for bar charts. Scale bars, (**a**) 50 ms, 5 nA; (**a,b**) 50 ms, 200 pA; (**d–f**) 50 ms, 500 pA.

## Discussion

We found that endogenous opioids released by electrical stimulation, activated a post-synaptic potassium conductance in Im neurons and also inhibited their glutamatergic and GABAergic synaptic inputs. Inhibiting peptidase activity potentiated these post-synaptic and pre-synaptic actions of the endogenously released opioid. At the BLA-Im synapse, endogenously released opioids inhibited neurotransmitter release through DOR but not MOR, even though both receptors are functional at the synapse, suggesting other factors control endogenous opioid action. A DOR PAM potentiated this inhibition and to our knowledge, this is the first example of an opioid PAM potentiating the actions of an endogenously released opioid. Up until now, endogenous opioids have been considered neuromodulators that regulate synapses and their plasticity during intense neuronal activity, which suggests an important role in learning[12,14–18]. These current findings indicate a different role of endogenously released opioids, namely as neuromodulators that are engaged by synaptic

activity to influence moment-to-moment information flow through dynamic Im-associated circuits.

It is likely the endogenous opioids regulating Im neuronal activity are met- and leu-enkephalin. The endogenous opioid actions we have described are through MOR and DOR, both of which are potently and efficaciously activated by the enkephalins[6,7]. We describe this phenomenon in Im neurons of the amygdala, which express high levels of enkephalins[20,21] and the effects are terminated by peptidases, known to breakdown enkephalins[38]. β-Endorphin is the other major opioid peptide that acts at MOR and DOR but is expressed in Im neurons at much lower levels[48] and is much less sensitive to peptidase degradation compared to enkephalin[49]. Although Im neurons have a low firing frequency *in vivo* ($<0.1\,Hz$)[29], subsequent orthodromic spike bursts[29] or recurrent firing[50] following synaptic stimulation or current injection could effectively release endogenous opioids. Indeed, our present findings support the premise that endogenous opioid release is activity dependent, however the intensity of this activity is much less than originally anticipated. Thus, within our working model (Fig. 7a), enkephalin is released in response to neuronal activity and this is likely from Im neurons, although it is possible that enkephalins are also released from BLA or CeA synaptic projections rather than from Im neurons alone.

Exogenously applied opioids activate a postsynaptic potassium conductance, in many neurons, including the Im[16,46,47]. However, while exogenously applied agonists inform us of what to expect when individuals are administered an opioid drug such as morphine, this method provides limited insight into how the endogenous opioid system functions. This study is the first example of endogenously released opioids, or in fact any endogenously released neuropeptide, activating a potassium conductance. The resulting membrane hyperpolarization would be expected to increase the excitatory synaptic input required to produce postsynaptic action potentials. In addition, the lower membrane resistance would likely shunt synaptic inputs from more distal synapses and thus could reduce the ability of all excitatory synaptic inputs to produce postsynaptic action potentials (Fig. 7a). This scenario would be particularly important if enkephalins are acting at the large dendrites we observed to be associated with ME immunereactive terminals (Fig. 4a). These additional roles for endogenous opioids as regulators of postsynaptic excitability and membrane resistance would be expected to have a widespread effect on information flow through the Im. Indeed, it is feasible that endogenously released opioids could limit the impact of all synaptic inputs on the membrane potential of Im neurons and thus alter feed-forward signalling in this neural circuit. This is in contrast to previous studies that describe endogenous opioid actions that are limited to inhibiting pre-synaptic inputs that express opioid receptors[13,14]. This is the first instance in which endogenous opioids have been shown to influence postsynaptic neuronal excitability, which in turn would be expected to alter the neuronal response to synaptic inputs.

In Im, we found that endogenous opioids are released in response to electrical stimulation, although reducing their breakdown or enhancing DOR sensitivity was required to observe a cellular effect. We found this surprising, as enkephalin is stored in dense core vesicles[51] (Fig. 1c–e) and optimal neuropeptide release from dense core vesicles traditionally occurs with more intense, higher frequency stimulation[37,52]. Consistent with this, endogenous opioid actions at other synapses require intense stimulation paradigms[12,14–18]. It is possible that dense core vesicle release is differently regulated in Im neurons, perhaps through differences in release machinery or intracellular calcium handling. Alternatively, we may be able to measure a response to

the small amount of peptide released because Im neurons are highly sensitive to opioids and the local microarchitecture between release site and receptor is favourable. Our finding that exogenous ME acts at both MOR and DOR at BLA-Im synapses, while endogenously released opioids only acts through DOR, further suggests local microarchitecture may govern the actions of endogenously released opioids. Both met-enkephalin and leu-enkephalin have similar affinity for MOR and DOR[6,7], so a difference in affinity for the receptors cannot explain our findings. Rather, MOR receptors on BLA-Im synapses may be located farther from the endogenous opioid release site or be more closely associated with membrane bound peptidases[53], both of which would reduce the ME concentration at MOR. A similar difference in receptor activation between exogenous and endogenously released agonist, occurs in the striatum[13]. This, together with our findings, suggests we should be wary of assuming the effects of exogenous agonists are indicative of how endogenous agonists signal. Regardless of the reason for the high sensitivity, this combination of ready releasable endogenous opioids and their action at multiple sites, including direct effects via a potassium conductance and the inhibition of synaptic inputs, would predict that endogenously released opioids are strong regulators of Im activity.

Activation of MOR and DOR results in opposite behavioural fear responses with activation of DOR being anxiolytic[8] and activation of MOR being anxiogenic[10]. The cellular basis for these opposing actions of opioid receptors is likely complex and occurs at multiple sites. At a cellular level, we have shown there is anatomical specificity of the endogenous opioid action at MOR and DOR in the Im. We suggest this could provide a basis for understanding the opposite actions of MOR and DOR activity on anxiety. In the Im, endogenously released opioids act via DOR to reduce the strength of BLA synaptic inputs. While this reduced excitatory drive from the BLA could decrease Im outputs to target neurons, it is also possible that it could allow Im neurons to be more strongly influenced by other inputs, such as those from the cortex that carry more complex/contextual information[31,54]. Thus, a DOR-mediated shift in strength of synaptic inputs could contribute to DOR-mediated anxiolytic processes such as fear extinction (Fig. 7b). Distinct from this, the endogenous opioid actions at MOR directly inhibit Im neuronal excitability and through this, likely reduce their activation by all synaptic inputs. This in turn, would reduce Im-dependent GABA release onto target neurons such as the CeM[16] and disinhibit CeM output to promote fear learning. Thus endogenous opioids acting via postsynaptic MOR to reduce Im excitability, maybe a feature of MOR-mediated anxiogenic processes such as fear learning. This differential pharmacology of endogenously released opioids in the Im could have important consequences. First, the net effect of endogenous opioid actions could change if one receptor subtype was altered by a physiological[55], pathophysiological state or pharmacological treatment[56]. Second, it raises some interesting therapeutic possibilities. If the desired therapeutic strategy is to enhance endogenous opioid action at both DOR and MOR, this could be achieved by inhibition of peptidases[57]. Alternatively, if enhancing the activity of only one receptor is desired, a PAM, such as shown in this study, could be utilized[45].

The reasons why anxiety disorders manifest are not fully understood, although it is thought a disruption in the balance between opposing circuits responsible for interpretation of fearful stimuli, a process in which the amygdala is heavily involved, maybe key[58]. It is possible endogenous opioid signalling is one of the essential components that maintain balance within these 'interpretation circuits', particularly during times of stress. If so, individual variability in fear and anxiety may result from differential expression of enkephalin. In fact, in a subpopulation

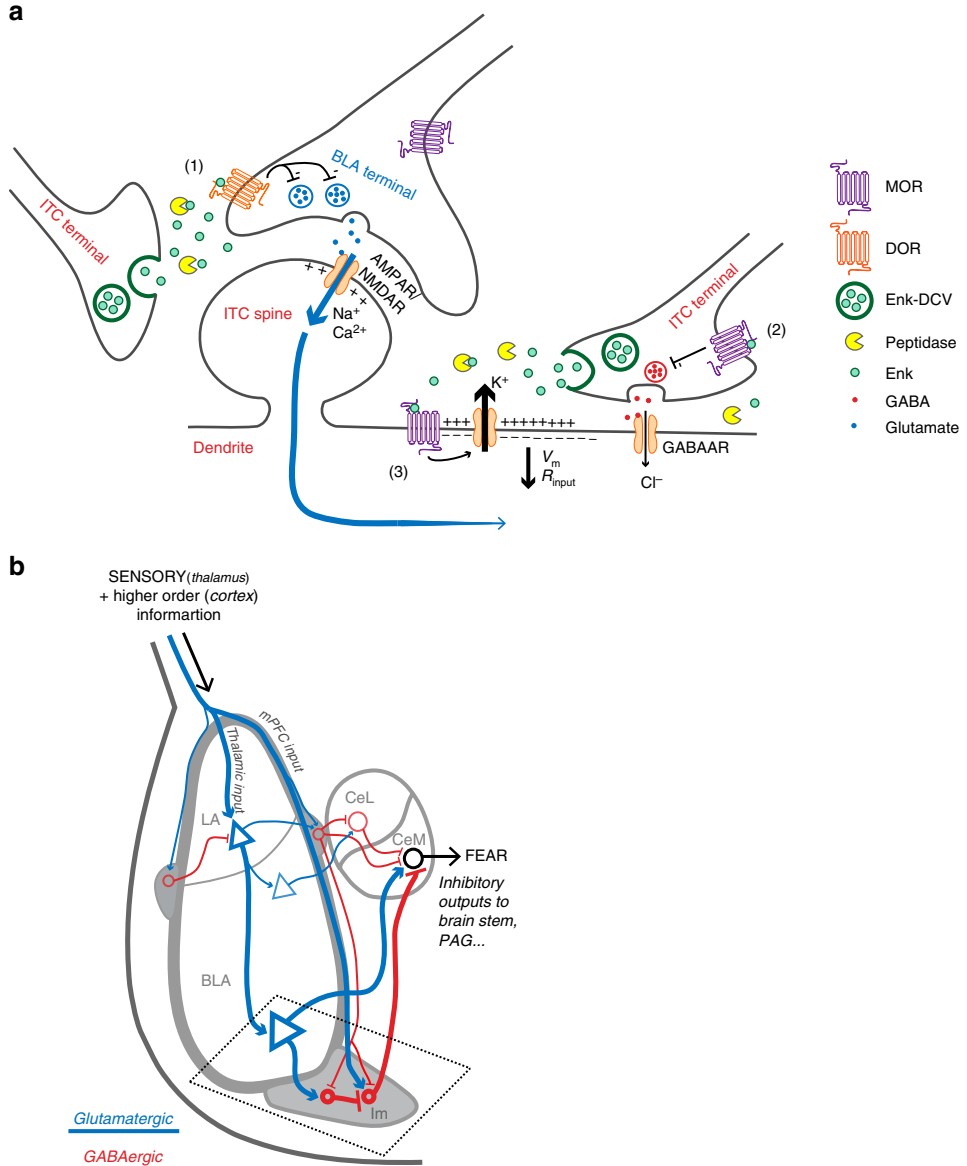

**Figure 7 | Working model of endogenous opioid actions in Im. (a)** Working model of endogenously released opioid signalling within Im. (1) Low–moderate stimuli at BLA-Im synapses promote release of endogenous enkephalins from dense core vesicles (DCV) contained within Im neurons. Sufficient peptide release through moderate stimulation is required to overcome peptidases and allow enkephalin signalling through DOR. Enkephalin reduces BLA-Im synaptic activity by decreasing pre-synaptic glutamate release. (2) Moderate stimuli, together with peptidase inhibition, are required to overcome potential microarchitectural constraints to allow enkephalin-induced MOR activation that reduces presynaptic GABA release at local Im–Im synapses. (3) Activation of postsynaptic MORs by endogenously released opioids activates a potassium conductance. The resulting efflux of K$^+$ ions hyperpolarizes ITCs reducing their excitability. Subsequent synaptic activity is also shunted (for example, blue arrows) due to decreased input resistance. Both outcomes are expected to reduce total Im activity and limit feed forward inhibition from Im to CeM neurons, thus affecting amygdala output. **(b)** Schematic representation of information flow through the amygdala. Putative thalamic/cortical glutamatergic (blue) afferents carry sensory information to the lateral amygdala (LA), activating BLA pyramidal neurons. These subsequently activate medial central amygdala (CeM) output neurons to promote fear behaviour. BLA excitatory afferents also project to Im, which then sends inhibitory GABAergic (red) efferents to the CeM. Other inputs (for example, from mPFC), traverse the intermediate capsule to activate Im neurons and contribute to feed-forward inhibition of CeM, preventing a fear response. Thinner lines depict putative inputs from the cortex, lateral/medial ITCs and the lateral central amygdala (CeL), all of which directly or indirectly affect amygdala output.

of rats where stress strongly reduces ME expression in the amygdala, this was correlated with increased vulnerability to negative stress responses[59]. While opioid receptors and peptides are expressed and likely act in other subdivisions of the amygdala[21,56,60], the strong regulation of Im neurons by endogenous opioids is an appealing mechanism for opioid regulation of fear and anxiety behaviours. Further, since endogenously released opioid regulation is engaged by moderate synaptic activity, this suggests some of their effects on anxiety maybe via regulation of continuous information flow through dynamic Im-associated neural circuits. This is in contrast to the currently perceived role in which endogenous opioids both regulate and are released by a high-intensity trigger such as a synaptic plasticity event or learning processes. In addition, endogenously released opioids may also regulate the expression of previously established plasticity/learning traces. This additional

role is important, as treatments for anxiety disorders are rarely able to preempt the initial learning but regulators of prior plasticity/learning could be valuable therapeutic options to reduce expression of fear and anxiety disorders.

## Methods

**Animals.** Brain tissue was prepared from male Sprague-Dawley rats (electrophysiology: 4–6 weeks; immunohistochemistry: 5–8 weeks; immuno-EM: adult), housed in standard environmental conditions with open topped cages, adequate enrichment, normal light/dark cycle (12 h/12 h) and *ad libitum* access to food and water. All experimental procedures were approved by either the Animal Care Ethics Committee of the University of Sydney or the Oregon Health & Science University IACUC and were conducted in accordance with either the Australian code of practice for the care and use of animals for scientific purposes or National Institutes of Health Guide for Care and Use of Laboratory Animals.

**Acute brain slice preparation.** Rats were anaesthetized with isoflurane, decapitated and their brains quickly removed and chilled in ice-cold cutting solution (in mM: NaCl, 125; NaHCO$_3$, 25; D-glucose, 11; KCl, 2.5; NaH$_2$PO$_4$.2H$_2$O, 1.25; MgCl$_2$, 2.5; CaCl$_2$, 0.5) saturated with carbogen (95% O$_2$/5% CO$_2$). Coronal slices (280 µm) containing the amygdala were cut with a vibratome (Leica) and incubated for 1 h at 34 °C then allowed to equilibrate to room temperature prior to recording. For recording, slices were transferred to a recording chamber superfused at 2 ml min$^{-1}$ with artificial CSF (aCSF) containing (in mM) NaCl, 125; NaHCO$_3$, 25; D-glucose, 11; KCl, 2.5; NaH$_2$PO$_4$.2H$_2$O, 1.25; MgCl$_2$, 1; CaCl$_2$, 2; saturated with carbogen and heated to 32–34 °C. Intercalated cells were visualized using an Olympus BX51 microscope equipped with × 40 water immersion objective and Dodt gradient contrast optics.

**Single whole-cell voltage-clamp recordings.** Main island ITCs (Im) were readily identified by their location, small cell body and dense population. To verify correct targeting of ITCs, slices were routinely fixed after recording and kept for *post hoc* staining (see below). Whole-cell voltage-clamp recordings were made from ITCs clamped at − 70 mV using patch pipettes (3–5 MΩ) containing (in mM): CsCl, 140; EGTA, 10; HEPES, 5; CaCl$_2$, 2; Mg-ATP, 2; Na-GTP, 0.3; QX314-Cl; 3 and 0.1% biocytin, pH 7.3, 280–285 mOsm l$^{-1}$. Electrically evoked synaptic responses were elicited via a bipolar tungsten stimulating electrode placed in one of two positions depending on the synaptic input studied (rate = 0.066 Hz unless otherwise stated, stimuli: 2–20 V, 100 µs). To record evoked excitatory postsynaptic currents (eEPSCs) at BLA-ITC synapses, the electrode was placed close to the basomedial edge of the BLA (Fig. 2b) and the GABA$_A$ receptor antagonists picrotoxin (100 µM, Sigma) and SR95531 (10 µM, Abcam Biochemicals) were included in the circulating aCSF to block fast inhibitory transmission. To record locally evoked inhibitory postsynaptic currents (eIPSCs), the electrode was placed within the main ITC cluster (Fig. 6e) and CNQX (10 µM, Tocris) or NBQX (5 µM, Abcam Biochemicals) were used to block fast excitatory synaptic transmission. Recordings of postsynaptic conductance were performed using whole-cell patch recordings in voltage-clamp mode holding at − 63.6 mV (adjusted for liquid junction potential measured at 13.6 mV). The pipette internal solution contained (in mM): potassium gluconate, 135; NaCl, 8; Mg-ATP, 2; Na-GTP, 0.3; EGTA, 0.5; HEPES, 10; pH 7.3, 280–285 mOsm l$^{-1}$. Continuous current recordings were monitored in chart mode and outward currents were calculated as the difference between baseline current and peak current during drug application. During stimulation, electrical stimuli were delivered to the BLA (10–20 pulses, 150 Hz) every 15 s and continuous current was recorded for 1.5 s after the stimulus train. To determine the reversal potential of exogenous ME-induced current, current–voltage analysis to sequential − 10 mV voltage steps from − 63.6 to − 133.6 mV (adjusted for liquid junction potential) was performed before and during ME application.

**Paired recordings of synaptically coupled Im neurons.** For paired recordings, designated pre-synaptic cells were loose-cell attached (seal resistance, 10–120 MΩ) and stimulated every 15 s with paired suprathreshold voltage pulses (200 mV, 50 ms interval) delivered via a patch pipette (2–3 MΩ) containing either aCSF or the potassium gluconate-based internal solution. Resulting IPSCs were recorded in post-synaptic cells that were whole-cell voltage clamped at − 70 mV with patch pipettes containing the CsCl-based internal solution (as above). Successful pairs were characterized by the generation of short latency (< 2 ms) IPSCs in the postsynaptic neuron in response to voltage stimulation of the presynaptic neurons. Synaptic failure was defined as complete failure to respond to both paired pulses and was calculated as a percentage of total episodes between the last 2.5 min of baseline and last 2.5 min drug application.

**Electrophysiology data acquisition and analysis.** Recordings were amplified, low-pass filtered (5 kHz), digitized and acquired (sampled at 10 kHz) using Multiclamp 700B amplifier (Molecular Devices) and online/offline analysis was with Axograph Acquisition software (Molecular Devices). In all cases, series resistance was monitored and data were discarded if this fluctuated more than 20% or if series

resistance > 20 MΩ. All postsynaptic currents were analysed with respect to peak amplitude. PPR was calculated from postsynaptic currents elicited by paired pulses (50 ms interval, second pulse/first pulse). Peak amplitude was quantified across each experiment as the mean peak amplitude of five to six episodes calculated from (1) the last 2–3 min of drug superfusion (that is, once the postsynaptic current had reached a stable plateau) and (2) the last 2–3 min of the baseline (referred to as CTL in figures), unless otherwise stated. Representative time plots of postsynaptic currents are shown as an average of two consecutive episodes (that is, every 30 s) with peak amplitude normalized to baseline averages. Numbers on bar charts represent 'n' = number of cells = number of slices. Effects of exogenously applied opioid agonists/antagonists were represented as percentage inhibition that reflects the difference between mean amplitude during agonist/antagonist superfusion and either baseline (CTL) or the preceding antagonist condition. Reversal (%) of the ME effect by selective antagonists was calculated as the proportion of amplitude recovered by the antagonist over total decrease in amplitude during ME. Endogenous opioid signalling was reflected by an increase in mean peak amplitude following either broad spectrum or selective opioid antagonist superfusion. Antagonist-induced increases (%) were calculated as the difference between amplitudes measured 7.5–10 min after antagonist superfusion and amplitudes at either baseline or the last 2.5 min of peptidase inhibitor superfusion (for example, Fig. 3a). In a subset of experiments, 10–90% rise time and synaptic decay of eIPSCs was analysed. The decay time course was fit to a double exponential function: $f(t) = A_1e(-t/\tau_1) + A_2e(-t/\tau_2)$ where $t$ is time, $A_1/A_2$ are peak amplitudes of the fast/slow decay components at $t = 0$ and $\tau_1/\tau_2$ are the fast/slow decay time constants, respectively. From this we calculated the weighted decay time constant ($\tau_w$) defined by: $\tau_w = (A_1\tau_1)/(A_1 + A_2) + (A_2\tau_2)/(A_1 + A_2)$.

**Drugs.** All drugs were diluted to their final concentration in aCSF and applied by superfusion. For experiments requiring the use of selective opioid agonists/antagonists, these drugs were applied cumulatively and sequentially, with the opioid receptor agonist/antagonist order combination varying between experiments to avoid bias. Concentrations of: naloxone (nalox; 10 µM), deltorphin II (delt; 300 nM), ICI174864 (ICI; 1 µM), CTAP (1 µM), naltrindole (nal, 10 nM) (all purchased from Tocris), DAMGO (1 µM), norBNI (10 nM), U69593 (U69; 3 µM) thiorphan (10 µM) (all Abcam Biochemicals), bestatin (10 µM, Cayman Chemicals) and captopril (1 µM, Sigma) remained the same for all experiments, while met-enkephalin (ME, Bachem; 100 nM, 300 nM, 500 nM, 10 µM, 30 µM) concentration was varied where indicated. BMS-986187 (1 µM) was from Bristol-Myer Squibb Co.

**Antibodies.** Antibodies used were as follows: ME (1:250, Millipore ab5026, RRID: AB_91644), MOR (1:5,000, Aves Labs), Alexa Fluor 488 anti-rabbit (1:800, ThermoFisher Scientific A21206, RRID: AB_10049650), Alexa Fluor 546 anti-chicken (1:500, ThermoFisher Scientific A11040, RRID: AB_2534097), Alexa Fluor 647 streptavidin (1:2,000, ThermoFisher Scientific S32357), DAPI (1:1,000, Thermo-Fisher Scientific 62248), biotinylated goat anti-rabbit IgG (1:400, Vector Laboratories BA-1000, RRID: AB_2313606), donkey anti-rabbit conjugated to gold particles (1 nm, 1:50, EMS 25700).

**Perfusion and slice preparation.** Animals (immunohistochemistry: $n = 6$; immuno-EM: $n = 3$) were deeply anaesthetized with isoflurane and/or lethal dose of pentobarbital sodium (120–150 mg kg$^{-1}$, i.p.). Once reflexes were abolished, animals were perfused through the ascending aorta with differing solutions depending on the imaging required. For confocal imaging animals received 3,000 units per ml heparin in a 0.5% NaNO$_2$/0.9% saline solution followed by 4% paraformaldehyde (PFA) solution in 0.1 M phosphate-buffered saline (PBS, pH 7.4). Brains were removed and post-fixed overnight in 4% PFA (4 °C), then washed three times with PBS and stored in PBS (4 °C) until sectioned. Coronal sections (30 µm) containing the amygdala were cut using a vibratome (Leica), and collected free-floating in PBS. Sections were stored in 40% PBS, 30% glycerol, 30% ethylene glycol cryoprotectant at − 30 °C until required for immunohistochemistry. For electron microscopy (EM), animals received the following sequence of solutions: (1) 10 ml of heparinized saline (1,000 units per ml); (2) 50 ml of 3.8% acrolein in 2% paraformaldehyde; and (3) 200 ml of 2% paraformaldehyde (in 0.1 M phosphate buffer (PB), pH 7.4). The block of brain containing the amygdala was placed in 2% paraformaldehyde for 30 min then into 0.1 M PB. Coronal sections (40 µm) were cut on a vibratome (Leica) and collected into 0.1 M PB. Prior to immunohistochemical processing, all sections were incubated in 1% sodium borohydride solution for 30 min to increase the antigenicity and 0.5% bovine serum albumin (BSA) for 30 min to reduce non-specific binding.

**Immunohistochemistry and confocal imaging.** Prior to staining, cryoprotectant was removed (3 × 10 min wash, PBS) then incubated 1 h at room temperature in 10% goat serum, 0.5% BSA and 0.3% Triton X-100 in PBS. Primary antibodies for ME and MOR were diluted in a 5% goat serum/0.3% Triton X-100 in PBS and sections were incubated overnight (4 °C). Secondary antibodies were diluted in 5% goat serum/0.3% Triton X-100 in PBS, and incubated 2 h at room temperature (light protected). The nuclear stain was added for the final 30 min of this

incubation. Sections were then washed (3 × 10 min) in PBS and mounted onto slides using ProLong Gold Antifade (Life Technologies).

For *post hoc* staining, slices (280 μm) containing cells filled with 0.1% biocytin (Sigma) during whole-cell recordings were fixed overnight at 4 °C in 4% PFA in 0.1 M PB, then washed three times with PB and stored (<2 weeks) at 4 °C prior to staining. For staining, slices were briefly washed before incubated for 1 h at room temperature in 10% goat serum, 0.5% BSA and 0.3% Triton X-100 in PB. Steptavidin conjugated antibody was diluted in 1% BSA/0.1% Triton X-100 in PB and incubated for 2 h at room temperature (light protected). DAPI was added for the final 30 min of this incubation period. Slices were then washed three to four times (10 min) with PB and mounted onto slides using Fluoromount-G (SouthernBiotech).

Sections were visualized using a Zeiss LSM510 Meta confocal microscope (lasers: 405, 488, 561 and 633 nm; Carl Zeiss) and Zeiss LSM META software. Images were taken sequentially with different lasers using × 10 (numerical aperture (NA) 0.45) and × 20 (NA 0.8) dry objectives, and × 63 (NA 1.4) oil immersion objective. Single images were collected using the × 10 objective and Z-stacks were collected at 4.5 and 2.5 μm for × 20 and × 63 oil objectives, respectively.

**Immuno EM labelling.** Alternate adjacent tissue sections from each animal were processed for single-labelling of ME utilizing either an immunoperoxidase or immunogold method. Sections were immersed in cryoprotectant solution (25% sucrose, 3% glycerol in 0.05 M PB) for 30 min and then briefly immersed in Freon followed by liquid nitrogen. This 'freeze-thaw' method increases penetration of antibodies into the surface of the tissue with a minimal disruption of morphology[61,62]. Tissue sections were incubated in a solution containing ME antibody for two nights at 4 °C. For visualization utilizing the avidin–biotin detection method[63] sections were incubated with a biotinylated IgG for 30 min followed by 30 min in avidin–biotin complex solution (Vector Laboratories) and 2 min in a DAB-hydrogen peroxide solution. For visualization by immunogold, these sections were incubated with secondary antibody conjugated to gold particles. Sections were rinsed in citrate buffer and colloidal gold particles were enhanced by silver enhancement for 6 min 30 s using the IntenSEM kit (GE Healthcare Life Sciences, Pittsburgh, PA). All incubations, except the primary antibody incubation, were carried out at room temperature with continuous agitation on a shaker table and sections were rinsed between incubations in the appropriate buffer (either 0.1 M Tris-saline or 0.1 M PB). The primary antibody incubation buffer also contained 0.1% BSA.

**Electron microscopy.** Following the immunoperoxidase or immunogold procedure, tissue sections were fixed for 15 min in 1.0% osmium tetroxide in 0.1 M PB, washed for 10 min in 0.1 M PB, dehydrated through a graded series of ethanols, placed into propylene oxide, and then a propylene oxide:EMBed (1:1) solution overnight. Sections were then incubated in EMBed for 2 h, embedded between two sheets of Aclar plastic, and placed in an oven for 24 h at 60 °C. Regions of amygdala were glued to plastic blocks formed in Beem capsules and ultrathin sections (75 nm) from an area just below the surface of the tissue at the tissue/plastic interface, where the penetration of antibodies is optimal were collected onto copper grids. The thin sections were counterstained with uranyl acetate and Reynolds lead citrate and examined using an FEI Tecnai 12 electron microscope (Hillsboro, OR). Immunoreactive (ir) profiles were captured using an AMT camera (Woburn, MA).

**Cell lines.** Chinese Hamster Ovary (CHO) PathHunter cells expressing enzyme acceptor (EA)-tagged β-arrestin 2 and ProLink (PK)-tagged DOR (CHO-OPRD1) were from DiscoveRx (Freemont, CA). Cells were grown in F-12 media (Invitrogen 11765), containing Hyclone FBS 10%, hygromycin 300 μg ml$^{-1}$ (Invitrogen 10687), G418 800 μg ml$^{-1}$ (Invitrogen 10131) and maintained at 37 °C in a humidified incubator containing 5% $CO_2$. These cells were used for β-arrestin recruitment assays and inhibition of forskolin-stimulated cAMP accumulation assays described below.

**PathHunter β-arrestin assay.** Confluent flasks of CHO-OPRD1 cells were collected with TrypLE Express, and resuspended in F-12 media supplemented with 10% FBS and 25 mM HEPES, at a density of $6.67e^5$ cells per ml and plated (3 μl per well) into white solid TC-treated 1,536-well plates (Corning, NY). Plates were incubated overnight at 37 °C in a 5% $CO_2$-humidified incubator. The next day, 1 μl of increasing concentrations of naloxone (4 × final concentration in assay buffer) were added to separate columns of the assay plates containing cells. Next, increasing concentrations of BMS-986187 (40 nl of 100 × final concentration in 100% DMSO) were added to separate rows of the assay plates by acoustic dispense using an Echo-550 (Labcyte, Sunnyvale, CA) from Echo-qualified 1,536-well source plates (Labcyte). Plates were covered with a lid and incubated at room temperature for 90 min. Incubations were terminated by the addition of 2 μl PathHunter Reagent. One hour later luminescence was detected using a Viewlux imaging plate reader (PerkinElmer). All PathHunter detection reagents were purchased from DiscoveRx, other chemicals, unless otherwise stated, were from Sigma.

**Inhibition of forskolin stimulated cAMP accumulation assays.** CHO-OPRD1 cells were grown to confluence then harvested and resuspended at $1e^6$ cells per ml in assay buffer (HBSS + 25 mM HEPES, + 0.05% BSA). Increasing concentrations of BMS-986187 (30 nl of 100 × final concentration in 100% DMSO) were added to separate rows of 1,536-well white solid NT plates by acoustic dispense using an Echo-550 (Labcyte, CA). Next, 1 μl of increasing concentrations of naloxone (at 3 × final concentration in assay buffer) were added to separate columns of the plates. Next, 1 μl of cells (1,000 cells per well) were added to all wells followed by 1 μl of forskolin (3 × final concentration in assay buffer). Plates were lidded and incubated for 45 min at RT. Incubations were terminated by the addition of Lance-Ultra cAMP detection reagent (1.5 μl of Eu-cryptate-labelled cAMP tracer in lysis buffer, followed by 1.5 μl of U-light-conjugated anti-cAMP antibody in lysis buffer). After a 1 h incubation at RT, time-resolved fluorescence was detected on a Viewlux or Envision plate reader (PerkinElmer) with excitation at 337 nm and emission reads at 615 and 665 nm. The ratiometric data (665 nm read/615 nm read) × 10,000 were converted to cAMP (nM) based on a standard curve for cAMP (replacing the cell addition step) run at the same time and under identical conditions to the assay. Lance-Ultra cAMP detection reagents were from PerkinElmer Life Sciences. All other chemicals, unless otherwise specified, were from Sigma.

**Statistical analysis.** No statistical methods were used to predetermine sample size, but our samples are similar to those generally used within the field. Data distribution was assumed normal in most cases except where sample size was $n < 5$, in which non-parametric tests were used; equal variance was also assumed although this was not formally tested. Results are reported as mean ± s.e.m. and two-sided statistical analysis was performed. Statistical significance was assessed with either parametric Student's *t*-test (paired or unpaired), repeated analysis of variance (ANOVA), one-way ANOVA, both with Tukey's *post hoc* adjustment or non-parametric Mann–Whitney U and Friedman's test with Fisher's *post hoc* adjustment, where appropriate. Statistical analysis was performed using SigmaPlot 10.0 (Systat Software) or SPSS 22.0 (IBM).

**Data availability.** The data that support the findings of this study are available from the corresponding author upon reasonable request.

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

## Acknowledgements

B.L.W., S.A.K., G.C.G., O.A.W., D.I.M. and E.E.B. were supported by National Health and Medical Research Council (NHMRC) project grants APP1047372, APP1077806, Bosch Institute Bishop Fellowship and USYD Thompson Fellowship. S.M.H. and S.A.A. were supported by National Institute of Health grant numbers R01 DE012640 and P30 NS061800. We thank Mark Connor, Macdonald Christie and Karen Aubrey for comments on the manuscript.

## Author contributions

This study was conceived by E.E.B. and designed by B.L.W. and E.E.B. Electrophysiological studies were conducted by B.L.W., S.A.K., G.C.G., D.I.M. and E.E.B. Immunohistochemical assays and confocal microscopy were conducted by O.A.W. and B.L.W. Immuno-EM and electron microscopy was conducted by S.M.H. and S.A.A. Biochemical assays were conducted by N.T.B. and A.A. The manuscript was written and edited by B.L.W. and E.E.B.

## Additional information

**Competing financial interests:** The authors declare no competing financial interests.

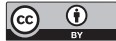

