## [Peer Review File · Nature Communications]

Reviewers' comments:

Reviewer #1 (Remarks to the Author):

The work in this manuscript has identified a site in the amygdala where the release of endogenous opioids results in a presynaptic inhibition of glutamate mediated by delta opioid receptors and a postsynaptic hyperpolarization mediated by mu opioid receptors. Two stimulus intensities were used to evoke the release. Blocking peptidase activity increased the amplitude and duration of the inhibition. The pharmacology was expertly done resulting in a convincing demonstration of the separate actions of the endogenous opioids. The results are convincing and conclusions are justified.

Comments

1. The work presented in this manuscript is exciting and important. Although there are a few examples of presynaptic inhibition mediated by endogenous opioids, there are only a few examples and the stimulus intensities necessary to release the opioids were intense. The work using lower intensity stimulation and the increase induced following the blockade of peptidases is very convincing. The activation of potassium conductance induced by endogenous opioids is the first example of its kind, is important and a long awaited observation.
2. There are a few details that need attention. The most important is in the presentation of the inhibition mediated by the release of opioids in Figures 3 & 4. It is difficult to see from the examples where the inhibition is and the reversal of the inhibition by naloxone were measured. It would be very helpful if the EPSC's that were used to make the bar graphs were indicated (maybe in gray boxes) in the example of the raw experiments.
3. The onset of inhibition induced by the peptidase inhibitors seemed to be quite variable. The long latency is particularly clear in Figure 4f and Figure 5. This long latency is quite different from the control experiments using low concentrations of ME in Figure 3e. It may be in some cases that an accumulation of endogenous opioids is necessary before a sufficiently high concentration is reached to activate the receptors. This seems to be the case particularly in Figure 5d.
4. The reversal induced by naloxone illustrated in Figure 4c & f is not complete and not super convincing.
5. There are small editorial issues. As a reader it would be helpful to spell out peptidase inhibitors and dense core vesicles throughout the manuscript. It would also be very helpful if the concentrations of compounds were given with reference to each application.

Reviewer #2 (Remarks to the Author):

The authors examined the modulation of ITC neurons by endogenous opioids using whole cell patch clamp recordings. Opioids suppressed BLA glutamatergic inputs to ITC cells and inhibitory transmission between ITC neurons. These two effects were mediated by distinct receptors, namely delta-opioid receptors (DOR) and mu-opioid receptors (MOR), respectively. Also, endogenous and exogenous opioids activated an outward current in ITC neurons. In addition, electron microscopic (EM) observations revealed appositions between met-enkephalin positive terminals and excitatory axon terminals or ITC dendrites.

Although the reduction of inter-ITC inhibition and the activation of the outward currents constitute a replication of an earlier study (Blaesse P, Goedecke L, Bazelot M, Capogna M, Pape HC, Jüngling K. J Neurosci. 2015 35:7317-25), the pre-synaptic inhibition of glutamatergic inputs to ITC cells as well as the EM observations are novel. However, I have several concerns with the experiments and the data presentation.

MAJOR

- 1) The authors claim that endogenous opioids continuously regulate synaptic inputs to ITC cells. In order to establish this, they would have to show that Naloxone alters synaptic transmission in baseline conditions. Instead, they show that naloxone is only effective when they stimulate the BLA repeatedly at high frequencies. However, this stimulation paradigm, by itself, likely altered synaptic transmission.
- 2) In the experiments assessing the effects of peptidase inhibitors (PIs), the authors excluded the cases where PIs enhanced synaptic transmission. Given that this study examines the impact of endogenous opioids, it is unclear why these cases were excluded?
- 3) In this manuscript, the authors conclude that GIRK is the K⁺ conductance activated by MORs. However no experiment was done to prove this. The authors did not even establish that a K⁺ conductance was involved.
- 4) The authors suggest that the differential modulation of glutamatergic and GABAergic transmission by DOR and MOR explains the opposite effects of DOR and MOR agonists on anxiety. However according to the results presented, activation of both receptors also suppresses the excitability of ITC cells. Thus, the contribution of opioids to anxiety regulation is likely much more complicated than suggested by the authors.
- 5) Instead of only showing representative examples, the authors should always illustrate averaged time course of %change in PSCs amplitudes.

MINOR

- 1) On page 11 lines 7-9, the authors note 'To test whether endogenously released opioids regulate GIRKs, we applied the PI cocktail and observed an outward current in some ITCs, which was fully reversed by CTAP or naloxone (Fig. 5c,e).' How many neurons were examined in these experiments? Also there is no trace for the naloxone condition in the figure.
- 2) In fig.6, instead of ICI, Naltrindole was used as a DOR antagonist. Please justify this change.
- 3) If, as the authors suggest, ITC cells are the source of endogenous opioids, they should fire spontaneously in the slice. However, no evidence of this was presented. For instance, were the effects of PI or BMS abolished by TTX?

Reviewer #3 (Remarks to the Author):

In this study the authors have examined the role of endogenous opioid modulation on synaptic activity in the main intercalated cell nucleus. Using whole cell recordings in acute brain slices combined with detailed anatomical analysis they show that in the main ITC cluster, endogenous release of enkaphalins, that activate opioid receptor have inhibitory actions at glutamatergic inputs from the basolateral amygdala. Interestingly they show that there are effects of both spontaneous and slow evoked release. While the action of exogenously applied opioids have been studied, to this is the first demonstration of actions of released opioids on synaptic transmission in a circuit that is thought to respond to endogenous opioids. This is a very interesting paper and has a very interesting result showing the actions of synaptically released agonist.

One concern is that in much of the text, they discuss the actions of opioids and other physiological actions on receptors in the intercalated nucleus. For example they indicate that the ITC nucleus is involved in fear extinction and also mention that BLA → ITC synapses are insensitive to regulation of MOR. Now, the ITC nuclei are a heterogeneous set of nuclei that surround the BLA that have been divided into the lateral, medial and main ITC clusters. These clusters have distinct connections, and most likely distinct functional roles. The above mentioned results pertain to the medial ITC nucleus, and as far as I know the main ITC nucleus has not been implicated in fear extinction and the actions of opioid receptor have also only been tested in the medial ITC nucleus. Little is known about the functional role and connectivity of the main ITC. They need to be much clearer as to what particular nuclei they are referring to throughout the manuscript. The difference in MOR modulation of BLA input

to the neurons studied here is a case in point as these inputs to the main ITC cluster clearly does not respond to the same receptors.

On the results on top of page 9, they refer to Fig 3e - this result actually seems to be Fig 3g? The data in this fig ((Fig 3) show rather small effects, and a wash would have been good as in Fig 2.

In Figure 6 they show that exogenous ME reduced evoked IPSCs and suggest that there is not change in the paired pulse ratio (PPR; Fig 6i). However, from the one cell shown in Fig 6b it seems apparent that the PPR appears to be changing. They suggest that the lack of change in PPR is due to complete block of one input. This is an important point as it raises the issue of which receptors are present where and what their mechanism of action is. They should show the PPR as the agonist goes on with time, or alternatively use submaximal doses of the agonist.

Minor: The introduction is not well written in my view and difficult to follow. For example the second paragraph, beginning with "Although opoid receptors " discusses connections of the ITC - however they treat the whole ITC clusters as one, and the message they are trying and they need to be clear on what they are referring to.

Reviewer #1 (Remarks to the Author):

We thank reviewer #1 for their positive feedback and have addressed their comments below:

1. *The work presented in this manuscript is exciting and important. Although there are a few examples of presynaptic inhibition mediated by endogenous opioids, there are only a few examples and the stimulus intensities necessary to release the opioids were intense. The work using lower intensity stimulation and the increase induced following the blockade of peptidases is very convincing. The activation of potassium conductance induced by endogenous opioids is the first example of its kind, is important and a long awaited observation.*

We appreciate the reviewer's interest in our work.

2. *"The most important is in the presentation of the inhibition mediated by the release of opioids in Figures 3 & 4. It is difficult to see from the examples where the inhibition is and the reversal of the inhibition by naloxone were measured. It would be very helpful if the EPSC's that were used to make the bar graphs were indicated (maybe in gray boxes) in the example of the raw experiments."*

As suggested we have now added grey boxes to the time plots to show where we measured the amplitude of the responses for calculations of drug-induced inhibition or potentiation (see Figures 2,3,4,6).

3. *"The onset of inhibition induced by the peptidase inhibitors seemed to be quite variable. The long latency is particularly clear in Figure 4f and Figure 5. This long latency is quite different from the control experiments using low concentrations of ME in Figure 3e. It may be in some cases that an accumulation of endogenous opioids is necessary before a sufficiently high concentration is reached to activate the receptors. This seems to be the case particularly in Figure 5d."*

We agree that the timing of the peptidase inhibitor effects is quite variable and it sometimes takes longer than application of low concentrations of ME. We also agree that this variability likely corresponds to the time taken to accumulate enough peptide to be able to measure an effect. We have addressed this more fully in the revised manuscript on page 9:

"It is important to note these targeted peptidases are non-selective metalloproteases. Therefore, blocking their activity has the potential to enhance the activity of other "off-target" endogenous signaling peptides (e.g. substance P^{39,40}, neurokinin^{41,42}; neurotensin^{42,43}) which may act either in concert or opposition to endogenous opioid signaling. Consistent with this, whilst inhibiting peptidase activity with the peptidase inhibitor cocktail on average, reduced eEPSC amplitude in response to both low ($12.9 \pm 5.4\%$ inhibition, n=7) and moderate ($32.1 \pm 6.2\%$ inhibition, n=8) stimuli (eg. Fig. 3g,h), there was a high degree of variability, with either inhibition (12/18 neurons); no effect (3/18 neurons) or an increase observed (3/18 neurons). Thus we defined endogenous opioid action as the naloxone-induced increase in eEPSC following peptidase inhibitor treatment (Fig. 3a). In addition, there was variability in the timing of the peptidase inhibitor response, which could reflect the time required to accumulate sufficient peptide or differences in restricting microarchitecture."

4. *"The reversal induced by naloxone illustrated in Figure 4c & f is not complete and not super convincing"*.

In regards to Fig. 4c, we would argue that whilst naloxone did not fully reverse the inhibition, on average it was within 10% of baseline levels. We have clarified this in the revised manuscript on page 10:

"BMS-986187 (1 μ M) significantly enhanced the inhibition induced by submaximal, exogenous ME (100 nM, Fig. 4c,e), indicating PAM activity at this synapse, which could be reversed

to baseline levels by naloxone ($6.3 \pm 2.6\%$ inhibition; $p > 0.05$ naloxone vs baseline, Tukey's *posthoc* comparisons, Fig. 4c,e)."

In regards to Fig. 4f, we hope the additional information provided in response to comment 3 (above and in revised manuscript) will address this comment. Specifically, it is possible that the peptidase inhibitor effect may result from multiple endogenous peptides acting in concert with endogenous opioids so that opioid antagonists are unable to completely reverse the effect.

5. *There are small editorial issues. As a reader it would be helpful to spell out peptidase inhibitors and dense core vesicles throughout the manuscript. It would also be very helpful if the concentrations of compounds were given with reference to each application.*

In the revised manuscript we have spelt out dense core vesicle and peptidase inhibitors throughout the manuscript. We have also added the drug concentration after the first use in the results for all drugs. In addition, as the met-enkephalin concentration differs between experiments the concentration is given for each application.

Reviewer #2 (Remarks to the Author)

We thank Review #2 for their feedback and comments and have addressed their comments below:

The authors examined the modulation of ITC neurons by endogenous opioids using whole cell patch clamp recordings. Opioids suppressed BLA glutamatergic inputs to ITC cells and inhibitory transmission between ITC neurons. These two effects were mediated by distinct receptors, namely delta-opioid receptors (DOR) and mu-opioid receptors (MOR), respectively. Also, endogenous and exogenous opioids activated an outward current in ITC neurons. In addition, electron microscopic (EM) observations revealed appositions between met-enkephalin positive terminals and excitatory axon terminals or ITC dendrites.

Although the reduction of inter-ITC inhibition and the activation of the outward currents constitute a replication of an earlier study (Blaesse P, Goedecke L, Bazelot M, Capogna M, Pape HC, Jüngling K. J Neurosci. 2015 35:7317-25), the pre-synaptic inhibition of glutamatergic inputs to ITC cells as well as the EM observations are novel. However, I have several concerns with the experiments and the data presentation.

We would like to highlight that whilst reviewer 2 indicates above that the effects we describe on 'inter-ITC inhibition and the activation of the outward currents' replicates previous work this is not the case. The earlier study, and our initial experiments, described exogenous opioid regulation of inter-ITC inhibition and outward currents. The big advance in this study is that we show that endogenously released opioids regulate inter-ITC inhibition, outward currents (and also BLA-Im synapses). This is not a replication of the earlier work and we highlighted this in the original manuscript, for example on page 15:

"Exogenously applied opioids activate post-synaptic GIRK in many neurons, including the Im^{16,46,47}. However, whilst exogenously applied agonists inform us of what to expect when individuals are administered an opioid drug such as morphine, this method provides limited insight into how the endogenous opioid system functions. This study is the first example of endogenously released opioids, or in fact any endogenously released neuropeptide, activating a potassium conductance, which is likely through GIRK activation."

MAJOR:

1. *"The authors claim that endogenous opioids continuously regulate synaptic inputs to ITC cells. In order to establish this, they would have to show that Naloxone alters synaptic transmission in*

baseline conditions. Instead, they show that naloxone is only effective when they stimulate the BLA repeatedly at high frequencies. However, this stimulation paradigm, by itself, likely altered synaptic transmission.”

We would like to stress that we do not think endogenous opioids are being released continuously in the absence of activity (as we indicated in our original manuscript ‘as neuromodulators engaged by basal synaptic activity’ in the abstract). Rather our data indicates that low levels of activity (Fig 3g, Fig 4d, Fig 5c,e), which would be expected within a dynamic circuit, are sufficient to induce endogenous opioid release. To reveal this effect we had to prevent peptidase activity or enhance receptor sensitivity, using a positive allosteric modulator. Perhaps the word basal is unclear or ambiguous and therefore we have changed our description to ‘neuromodulators engaged by minimal synaptic activity’ in the abstract and throughout and think this is entirely appropriate.

2. In the experiments assessing the effects of peptidase inhibitors (PIs), the authors excluded the cases where PIs enhanced synaptic transmission. Given that this study examines the impact of endogenous opioids, it is unclear why these cases were excluded?

As highlighted above, in response to reviewer 1 comment 3, we used the peptidase inhibitors as a tool to enhance endogenous opioid signalling by preventing peptidase-dependent breakdown. We have clarified why we excluded these cells in the revised manuscript on page 9:

“Despite this variability in peptidase inhibitor response, naloxone significantly increased eEPSC amplitudes in all cells. Due to possible dominance of confounding “off-target” signaling peptides in cells where the peptidase inhibitors increased the eEPSC amplitude, these cells were excluded from further analysis of the endogenous opioid effect.”

3. In this manuscript, the authors conclude that GIRK is the K⁺ conductance activated by MORs. However no experiment was done to prove this. The authors did not even establish that a K⁺ conductance was involved.

We have conducted further experiments to address this concern and the data is presented in figure 3c and is described in the text on page 11:

“Current-voltage analysis before and during ME indicates activation of a potassium conductance as the reversal potential of the ME-induced current was close to the potassium reversal potential (ME_{rev}: -103.2 ± 1.4 mV, n=9; E_k: -104.9 mV, calculated with the Nernst equation; Fig. 5c).”

4. “The authors suggest that the differential modulation of glutamatergic and GABAergic transmission by DOR and MOR explains the opposite effects of DOR and MOR agonists on anxiety. However according to the results presented, activation of both receptors also suppresses the excitability of ITC cells. Thus, the contribution of opioids to anxiety regulation is likely much more complicated than suggested by the authors.”

We agree with reviewer 2 that the contribution of endogenous opioids to anxiety regulation is complex and likely results from endogenous opioids acting at multiple sites. We have therefore strengthened our comments in this regard in the revised manuscript. However, we do not agree with their suggestion that both MOR and DOR activation will suppress the excitability of Im cells and thus produce similar effects. Rather, the cellular physiology in our paper has revealed a specific anatomic divide in endogenous opioids acting at MORs and DORs within Im. Whilst, direct post-synaptic inhibition of Im neurons was through MOR, inhibition in glutamate release from BLA terminals was through DOR. Thus, MOR receptor activation would be expected to inhibit Im neuronal excitability whereas DOR activation would not. Of course, inhibition of this single glutamatergic input could reduce excitatory drive to Im neurons. However, this is quite different to direct inhibition of Im neuron excitability and does not take into account other synaptic inputs, which may dominate in place of BLA-Im signaling. With this confusion about both receptors inhibiting Im neurons removed, the

opposing consequences of DOR or MOR activation we have suggested becomes feasible. Indeed, we propose the different functional consequences of endogenous opioids acting at these anatomically specific MOR and DOR sites may provide a basis for understanding the opposing actions of opioids in fear behaviours. As this was unclear in the original submission, we have changed our explanation in the revised manuscript on page 18:

“Activation of MOR and DOR results in opposite behavioral fear responses with activation of DOR being anxiolytic⁸ and activation of MOR being anxiogenic¹⁰. The cellular basis for these opposing actions of opioid receptors is likely to be complex and occur at multiple sites. At a cellular level, we have shown there is anatomical specificity of the endogenous opioid action at MOR and DOR in the Im. We suggest this could provide a basis for understanding the opposite actions of MOR and DOR activity on anxiety. In the Im, endogenously released opioids act via DOR to reduce the strength of BLA synaptic inputs. Whilst this reduced excitatory drive from the BLA could decrease Im outputs to target neurons, it is also possible that it could allow Im neurons to be more strongly influenced by other inputs, such as those from the cortex that carry more complex/contextual information^{31,54}. Thus, a DOR mediated shift in strength of synaptic inputs could contribute to DOR-mediated anxiolytic processes such as fear extinction (Fig. 7b). Distinct from this, the endogenous opioid actions at MOR directly inhibit Im neuronal excitability and through this likely reduce their activation by all synaptic inputs. This in turn, would reduce Im-dependent GABA release onto target neurons such as the CeM¹⁶ and disinhibit CeM output to promote fear learning. Thus endogenous opioids acting via postsynaptic MOR to reduce Im excitability, maybe a feature of MOR-mediated anxiogenic processes such as fear learning.”

5. *“Instead of only showing representative examples, the authors should always illustrate averaged time course of %change in PSCs amplitudes.”*

We presented example time plots from single cells because grouped time plots were not possible for two reasons. Firstly, due to experimental design as we chose to rotate drug order to ensure no bias (Fig 2c, 4a, as described in drug section of the methods) and secondly, because the timing of the endogenous opioid effect varied between experiments.

MINOR:

1. *“On page 11 lines 7-9, the authors note 'To test whether endogenously released opioids regulate GIRKs, we applied the PI cocktail and observed an outward current in some ITCs, which was fully reversed by CTAP or naloxone (Fig. 5c,e).' How many neurons were examined in these experiments? Also there is no trace for the naloxone condition in the figure.”*

We have clarified the antagonists used in these experiment on pages 11 and 12 of the revised manuscript. We chose an example trace using the selective mu opioid antagonist CTAP and did not include a trace with naloxone as the responses were the same.

2. *“In fig.6, instead of ICI, Naltrindole was used as a DOR antagonist. Please justify this change.”*

Unfortunately, it was necessary to change from ICI to Naltrindole for the locally stimulated Im synapses as ICI was unavailable at the time we conducted these experiments. We used supramaximal but still selective concentrations of both antagonists. Naltrindole was without effect but the ME response was completely reversed by the MOR selective antagonist CTAP (Fig 6h). This suggests that the ME effect at this synapse is only mediated by MOR and as a consequence naltrindole did produce an affect.

3. *“If, as the authors suggest, ITC cells are the source of endogenous opioids, they should fire spontaneously in the slice. However, no evidence of this was presented. For instance, were the effects of PI or BMS abolished by TTX?”*

Whilst it is possible that reducing all firing in the slice could eliminate endogenous opioid release we used the opposite approach to show that the endogenous opioid effect was dependent on the increased activity level of the slice. This highlights the activity-dependent nature of endogenous opioid release. We have added information about Im firing frequency into the revised manuscript on page 15:

“Although Im neurons have low firing frequency *in vivo* (<0.1 Hz)²⁹, subsequent orthodromic spike bursts²⁹ or recurrent firing⁵⁰ following synaptic stimulation or current injection could effectively release endogenous opioids.”

In addition in our original manuscript, we did not discount the possibility of endogenous opioids being released from alternative sites such as the BLA or CeA as stated on pages 15 and 16:

“Thus, within our working model (Fig. 7a), enkephalin is released in response to minimal activity and this is likely from Im neurons, although it is possible that enkephalins are also released from BLA or CeA synaptic projections rather than from Im neurons alone.”

Reviewer #3 (Remarks to the Author)

We thank reviewer #3 for their positive feedback and comments, which we have addressed below:

1. *“One concern is that in much of the text, they discuss the actions of opioids and other physiological actions on receptors in the intercalated nucleus. For example they indicate that the ITC nucleus is involved in fear extinction and also mention that BLA -> ITC synapses are insensitive to regulation of MOR. Now, the ITC nuclei are a heterogenous set of nuclei that surround the BLA that have been divided into the lateral, medial and main ITC clusters. These clusters have distinct connections, and most likely distinct functional roles. The above mentioned results pertain to the medial ITC nucleus, and as far as I know the main ITC nucleus has not been implicated in fear extinction and the actions of opioid receptor have also only been tested in the medial ITC nucleus. Little is known about the functional role and connectivity of the main ITC. They need to be much clearer as to what particular nuclei they are referring to throughout the manuscript. The difference in MOR modulation of BLA input to the neurons studied here is a case in point as these inputs to the main ITC cluster clearly does not respond to the same receptors.”*

We agree we should have been clearer on what is known about the Im versus other ITC clusters. We have altered the introduction on page 4 to address this:

“Opioid receptors and peptides are expressed to varying degrees throughout the amygdala^{20,21}. In particular, the intercalated cells (ITCs) are one possible site where enkephalin could regulate fear and anxiety behaviors. ITCs are small clusters of densely packed GABAergic neurons that en-sheath the basolateral amygdala (BLA). Coronal sections give rise to three separate clusters: the smaller lateral (lpc) and medial (mpc) paracapsular ITC clusters are located within the external and intermediate capsules respectively and the larger main island (Im), is located ventromedial to the BLA²² (Fig. 1a). Whilst lpcs provide feedforward inhibition to the BLA²³, mpcs act as an inhibitory interface between the BLA and CeA and thus regulate fear learning²⁴. In particular the mpcs are required for fear extinction²⁵. Less is known about the functional role of the main island although it is possible the Im plays a similar role to mpcs. Indeed, Im neurons also receive sensory information from both the BLA^{24,26} and the thalamus²⁷ along with more complex information from the medial pre-frontal cortex (mPFC), a region highly implicated in fear extinction^{28,29}. Like the mpcs the Im sends inhibitory GABAergic projections to the medial central nucleus (CeM)^{21,26,30} and thus could gate expression of the conditioned fear response³¹. The Im may be particularly important during fear extinction as Im neurons are activated by extinction^{22,32}, their ablation (along with mpcs) reduces extinction²⁵ and treatments that reverse the extinction deficit in anxious mice elevate Im neuron activity³².”

Opioid regulation of Im membrane currents and synaptic inputs from the BLA was indeed described in the Blaesse et al., (2015) study referred to by reviewer 3. The confusion arises because different groups use different names for the same ITC clusters. In the introduction of Blaesse et al., (J. Neurosci 35:7317-7325, 2015) it is made clear that the mITC(v) neurons are what is referred to by others as the Im; it states “dorsal (mITC(d)) and ventral (mITC(v)) groups; the latter is also referred to as the main ITC nucleus.” Therefore, the same population of ITC cells described by Blaesse include what we and others call the Im. To ensure this is clear on page 4 in the revised manuscript we have added:

“in Im neurons (Im referred to as medial ventral ITC in this study)¹⁶”

2. “On the results on top of page 9, they refer to Fig 3e - this result actually seems to be Fig 3g? The data in this fig ((Fig 3) show rather small effects, and a wash would have been good as in Fig 2.”

Reviewer three is correct, we were indeed referring to Fig. 3g and have corrected this in the text.

As clarified in response to comment 3 of reviewer 1 we defined endogenous opioid action as the naloxone-induced increase in eEPSC following peptidase inhibitor treatment. We have now added grey boxes to the time plots to show where we measured the amplitude of the responses for calculations of drug-induced inhibition or potentiation in figure 3. We hope this allows the reader to better evaluate the magnitude of the endogenous opioid effect and confirm that they align with the grouped data shown in Fig. 3i.

We are unsure which drug reviewer 3 suggests we wash from the slice. Washing peptidase inhibitors from the slice would not allow us to assess the opioid receptor induced component of the response and due to its slow dissociation rate, washing naloxone (Pasternak & Synder, Nature 253, 563-565, 1975) from a slice is not feasible.

3. “In Figure 6 they show that exogenous ME reduced evoked IPSCs and suggest that there is not a change in the paired pulse ratio (PPR; Fig 6i). However, from the one cell shown in Fig 6b it seems apparent that the PPR appears to be changing. They suggest that the lack of change in PPR is due to complete block of one input. This is an important point as it raises the issue of which receptors are present where and what their mechanism of action is. They should show the PPR as the agonist goes on with time, or alternatively use submaximal doses of the agonist.”

We would like to highlight that in Fig. 6i and Fig. 6b we are utilising two distinct recording methods, as indicated on pages 12-13. Synaptic pairs for Fig. 6a-d and local electrical stimulation for Fig.6e-k. Reviewer three is correct that we observe a change in paired pulse ratio in 6b when we are recording the GABAergic response from a neuron pair. This is quantified in 6d. However, when we changed to electrically stimulating within the Im nucleus, the change in PPR was not consistent and did not reach statistical significance (Fig. 6i). We have clarified this on page 13:

“Surprisingly however, we did not find a consistent change in PPR (Fig. 6i), which was in distinct contrast to opioid regulation of glutamatergic inputs (Fig. 2e) and paired Im neuron recordings (Fig. 6a-d).”

4. “The introduction is not well written in my view and difficult to follow. For example the second paragraph, beginning with “Although opioid receptors ” discusses connections of the ITC - however they treat the whole ITC clusters as one, and the message they are trying and they need to be clear on what they are referring to.”

We have revised the introduction to clarify the specific ITC clusters, and hope it is not clearer and easier to differentiate which ITC cluster we are talking about.

Reviewers' comments:

Reviewer #1 (Remarks to the Author):

The response to the comments and suggestions are appropriate and I am satisfied.

Reviewer #2 (Remarks to the Author):

The authors have addressed some of my comments but I am not satisfied with their responses to the following two comments.

1. In my first review, I wrote: "The authors claim that endogenous opioids continuously regulate synaptic inputs to ITC cells. In order to establish this, they would have to show that Naloxone alters synaptic transmission in baseline conditions. Instead, they show that naloxone is only effective when they stimulate the BLA repeatedly at high frequencies. However, this stimulation paradigm, by itself, likely altered synaptic transmission."

The authors replied: "We would like to stress that we do not think endogenous opioids are being released continuously in the absence of activity (as we indicated in our original manuscript 'as neuromodulators engaged by basal synaptic activity' in the abstract). Rather our data indicates that low levels of activity (Fig 3g, Fig 4d, Fig 5c,e), which would be expected within a dynamic circuit, are sufficient to induce endogenous opioid release. To reveal this effect we had to prevent peptidase activity or enhance receptor sensitivity, using a positive allosteric modulator. Perhaps the word basal is unclear or ambiguous and therefore we have changed our description to 'neuromodulators engaged by minimal synaptic activity' in the abstract and throughout and think this is entirely appropriate."

The problem remains. While opioids may be released by minimal synaptic activity, they have no effect in these circumstances unless peptidase activity is blocked, which of course does not occur in vivo. Strong stimulation paradigms are needed to show opioid effects. The authors must tone down all the statements about minimal synaptic activity.

3. In my first review, I wrote: "In this manuscript, the authors conclude that GIRK is the K⁺ conductance activated by MORs. However no experiment was done to prove this. The authors did not even establish that a K⁺ conductance was involved."

The authors replied: "We have conducted further experiments to address this concern and the data is presented in figure 3c and is described in the text on page 11:"

Unfortunately, the new experiments did not address my concern. Positive evidence that GIRK mediates the opioid effect is still lacking. For instance, measuring the I-V of the PI-induced current and using Tertiapin Q instead of CTAP in fig. 3d would address my concern. It should be noted that the current induced by ME does not seem to show inward-rectification, which is inconsistent with a mediation by GIRK.

Reviewer #3 (Remarks to the Author):

The authors have addressed all my concerns, The only issue I have is that, in Fig 6 it would be good to see the PPR plotted with time as is the synaptic current. However, this is a minor point. I congratulate the authors on a very interesting rest.

Response to reviewer 2

Reviewer #2 (Remarks to the Author):

The authors have addressed some of my comments but I am not satisfied with their responses to the following two comments.

1. In my first review, I wrote: "The authors claim that endogenous opioids continuously regulate synaptic inputs to ITC cells. In order to establish this, they would have to show that Naloxone alters synaptic transmission in baseline conditions. Instead, they show that naloxone is only effective when they stimulate the BLA repeatedly at high frequencies. However, this stimulation paradigm, by itself, likely altered synaptic transmission."

The authors replied: "We would like to stress that we do not think endogenous opioids are being released continuously in the absence of activity (as we indicated in our original manuscript 'as neuromodulators engaged by basal synaptic activity' in the abstract). Rather our data indicates that low levels of activity (Fig 3g, Fig 4d, Fig 5c,e), which would be expected within a dynamic circuit, are sufficient to induce endogenous opioid release. To reveal this effect we had to prevent peptidase activity or enhance receptor sensitivity, using a positive allosteric modulator. Perhaps the word basal is unclear or ambiguous and therefore we have changed our description to 'neuromodulators engaged by minimal synaptic activity' in the abstract and throughout and think this is entirely appropriate."

The problem remains. While opioids may be released by minimal synaptic activity, they have no effect in these circumstances unless peptidase activity is blocked, which of course does not occur in vivo. Strong stimulation paradigms are needed to show opioid effects. The authors must tone down all the statements about minimal synaptic activity.

We have toned down our statements about endogenous opioid action with minimal stimulation throughout the manuscript (please see tracked changes). However, in the discussion section about endogenous opioid **release** (rather than cellular action) we have left a comment that low stimulation seems to be sufficient for endogenous opioid release. We have added an additional comment to highlight that whilst the endogenous opioids may be released by low stimulation you cannot observe their actions under these conditions unless you reduce their breakdown or include a positive allosteric modulator. So we have stepped back from saying that endogenous opioids act under conditions of low stimulation but the evidence we have suggests they are released under these conditions. We think this distinction (which possibly wasn't clear enough previously) is important as it informs us about release of peptides (but possibly not their actions).

3. In my first review, I wrote: "In this manuscript, the authors conclude that GIRK is the K⁺ conductance activated by MORs. However no experiment was done to prove this. The authors did not even establish that a K⁺ conductance was involved."

The authors replied: "We have conducted further experiments to address this concern and the data is presented in figure 3c and is described in the text on page 11:"

Unfortunately, the new experiments did not address my concern. Positive evidence that GIRK mediates the opioid effect is still lacking. For instance, measuring the I-V of the PI-induced current and using Tertiapin Q instead of CTAP in fig. 3d would address my concern. It should be noted that the current induced by ME does not seem to show inward-rectification, which is inconsistent with a mediation by GIRK.

We have included further experiments that indicate that the opioid

effect is mediated by GIRK. These experiments show that the application of exogenous enkephalin is reduced by 83% by the GIRK blocker tertiapin Q (page 12 of revised manuscript). This data together with the IV showing a reversal potential near the potassium reversal potential provide very strong evidence that enkephalin activates a GIRK in intercalated cells. We chose to test whether the exogenous enkephalin current was sensitive to tertiapin Q rather than the endogenous opioid effect because the exogenous opioid action occurs with low variability in all intercalated cells. To perform the tertiapin Q experiment on the endogenous opioid current is technically very difficult and does not provide a significant meaningful advance. These experiments are technically very difficult because of the much longer experiment required to combine both tertiapin Q and the peptidase inhibitors and the much greater variability between cells of the endogenous opioid action (when compared to the exogenous drug).

The reviewer is correct that the ME-induced outward current does not display significant rectification. Given the sensitivity of this conductance to the GIRK 1/4 (Kir3.1/Kir3.4) inhibitor we are confident that the conductance ME activates is a GIRK and is largely due to channels that contain GIRK1 and GIRK4. GIRK1 has the smallest inward rectification of the GIRK family of channels and GIRK 4 is only expressed at low levels in the brain. The rectification of GIRK relies on intracellular blockade of the channel by Mg^{2+} or polyamines for outward ion flow. Therefore, it is possible that whole cell recording from these very small neurons may have dialysed these intracellular components more fully than in larger neurons. Therefore, even though we do not see rectification given the addition of the tertiapin Q data we are confident that the ME-induced current is a GIRK. However, the importance of these experiments does not rely on the ME-induced current being GIRK. Therefore, if this is not satisfactory evidence for the involvement of GIRK in your mind we would be happy to

call it a tertiapin Q sensitive potassium conductance.

REVIEWERS' COMMENTS:

Reviewer #1 (Remarks to the Author):

I was convinced the first time that this manuscript was submitted, the response to the first set of comments and suggestions improved it and I remain very enthusiastic about it.

Reviewer #3 (Remarks to the Author):

With regard to the concern about Ref #2 first concern.

I agree with the reviewers that while there is apparent evidence that opioids can be released with basal activity in a slice – it has no effect without peptidase blockers on board. Whether these are released and active in any way in vivo now becomes a moot point. Moreover to the response to Comment #3 about the nature of the current – the authors have stated that using tertiapam Q to block the 'endogenous opioid current' is difficult because of the "much greater variability between cells of the endogenous opioid action". This calls into question the entire finding regarding the physiological impact of the endogenous opioid current.

With regard to question #3 – the evidence that the current is a GIRK current. The nature of the current evoked by exogenous opioid is blocked by Tert Q – however it is not showing inward rectification. They suggest that intracellular polyamines are washing out because these are small neurons. The cell body of these cells is small but the recent fills from the Luthi lab show that they are rather large – moreover, GIRK current were first identified in pyramidal neurons where the washout should be more efficient. Thus whether this is a GIRK current is still not clear and a Tert sensitive K current is not realistic. The question really is related to the endogenous current – for this current – it is not clear that the same current is active. The evidence entirely hinges on the pharmacology.

REVIEWERS' COMMENTS:

Reviewer #1 (Remarks to the Author):

I was convinced the first time that this manuscript was submitted, the response to the first set of comments and suggestions improved it and I remain very enthusiastic about it.

Reviewer #3 (Remarks to the Author):

With regard to the concern about Ref #2 first concern.

I agree with the reviewers that while there is apparent evidence that opioids can be released with basal activity in a slice – it has no effect without peptidase blockers on board. Whether these are released and active in any way in vivo now becomes a moot point. Moreover to the response to Comment #3 about the nature of the current – the authors have stated that using tertiapan Q to block the ‘endogenous current’ is difficult because of the “ much greater variability between cells of the endogenous opioid action “. This calls into question the entire finding regarding the physiological impact of the endogenous opioid current.

We have removed any reference to minimal synaptic stimulation throughout the manuscript. In particular:

Abstract line 19: removal of ‘minimal’

Abstract line 26: removal of ‘minimal’

Line 93: removal of ‘require no or minimal external stimulation and’

Line 329: removed ‘minimal’

Line 340: removed ‘regular minimal’

Line 358: removed ‘minimal’

Line 382: removed ‘no or low levels of’

Line 398: removed ‘in response to low stimulation’

Line 450: replaced ‘minimal’ with ‘moderate’

With regard to question #3 – the evidence that the current is a GIRK current. The nature of the current evoked by exogenous opioid is blocked by Tert Q – however it is not showing inward rectification. They suggest that intracellular polyamines are washing out because these are small neurons. The cell body of these cells is small but the recent fills from the Luthi lab show that they are rather large – moreover, GIRK current were first identified in pyramidal neurons where the washout should be more efficient. Thus whether this is a GIRK current is still not clear and a Tert sensitive K current is not realistic. The question really is related to the endogenous current – for this current – it is not clear that the same current is active. The evidence entirely hinges on the pharmacology.

We have removed any reference to the conductance being GIRK. In particular:

Line 79: replace ‘G-protein coupled inwardly rectifying potassium conductance (GIRK)’ with ‘potassium conductance’

Line 245: replace ‘GIRKs’ with ‘a potassium conductance’

Lines 245-246: replace ‘GIRK’ with ‘this potassium conductance’

Line 267: replace ‘GIRKs’ with ‘a potassium conductance’

Line 277: deleted ‘likely through activation of GIRKs’

Line 362: replaced ‘through GIRKs’ with ‘potassium conductance’

Line 367: deleted ‘which is likely through GIRK activation’

Line 411: replaced ‘GIRK’ with potassium conductance

Line 1000-1001: replace ‘GIRK channels’ with potassium conductance

Line 1001: replace ‘GIRK’ with ‘the potassium conductance’